# Sequential temporal anticipation characterized by neural power modulation and in recurrent neural networks

Xiangbin Teng[1,2]*, Ru-Yuan Zhang[3]*

[1]Department of Psychology, The Chinese University of Hong Kong, Hong Kong SAR, China; [2]Brain and Mind Institute, The Chinese University of Hong Kong, Hong Kong SAR, China; [3]Brain Health Institute, National Center for Mental Disorders, Shanghai Mental Health Center, Shanghai Jiao Tong University School of Medicine and School of Psychology, Shanghai, China

## eLife assessment

This **valuable** study provides insights into how the brain learns to better detect a target by predicting when the target may appear. Overall, **solid** evidence is provided that the power fluctuations of alpha- and beta-band oscillations can reflect the predicted occurrence time of the target, but some conclusions, especially ones related to the neural-network model and temporal gain control account, need further consideration. The study highlights an advanced EEG analysis approach as well as a close combination of human EEG analysis and computational modeling using recurrent neural networks.

*For correspondence:
XiangbinTeng@cuhk.edu.hk (XT);
ruyuanzhang@sjtu.edu.cn (R-YZ)

**Competing interest:** The authors declare that no competing interests exist.

**Abstract** Relevant prospective moments arise intermittently, while most of the time is filled with irrelevant events, or noise, that constantly bombard our sensory systems. Thus, anticipating a few key moments necessitates disregarding what lies between the present and the future – the noise. Here, through examining how the brain and recurrent neural networks (RNNs) anticipate a sequence of prospective moments without relying on any external timing cues, we provided a reinterpretation of temporal anticipation. We first designed a 'premembering' task, where we marked three temporal locations in white noise and asked human listeners to detect a tone at one of these points. Using power modulation analyses, we investigated the memory-guided anticipatory processes in trials involving only flat noise. Our research revealed a unique neural-power modulation pattern for sequential temporal anticipation: neural power within the alpha-beta band range fluctuates over time, accurately identifying the marked locations on a sub-second scale and correlating with tone detection performance. To understand the functional roles of these neural modulations, we utilized RNNs optimized for the behavioral task. The hidden dynamics of the RNNs mirrored the neural modulations, and additional analyses and perturbations on the RNNs indicated that the neural power modulations in the alpha-beta band resulted from selective suppression of irrelevant noise periods and heightened sensitivity to anticipated temporal locations. Our behavioral, neural, and modeling findings collectively indicate that sequential temporal anticipation involves a process of dynamic gain control: anticipating a few significant moments involves actively disregarding irrelevant events that frequently occur.

## Introduction

In a common laboratory setup for studying temporal anticipation, a subject - either an animal or a human - is tasked with predicting a brief sensory target, such as a tone or visual pattern. These targets usually last for only a few hundred milliseconds or less but are embedded within a much longer sequence of stimuli, noise, or silence that can last for seconds. This scenario also happens in our daily lives. For instance, we estimate the moment a traffic light will turn green to promptly start our car, despite waiting for minutes at a large intersection. In both the laboratory setting and daily life, the key prospective moments are akin to small islands amidst a sea of irrelevant events or noise. Thus, the role of 'temporal anticipation' likely extends beyond merely predicting a signal; it also involves ignoring the 'noise' that exists between the present and future. In this study, we explore sequential temporal anticipation to test how both human participants and recurrent neural networks (RNNs) predict future events in noisy environments.

An ecological task that the brain solves is to anticipate several time points sequentially – 'sequential temporal anticipation' such as in communicating speech and music (*Nobre and van Ede, 2018*; *Rimmele et al., 2018*). Take the example of learning to play the piano. While the musical notes in a melody are played one after the other at specific intervals, piano players plan a sequence of actions, which necessitates their brain to anticipate a series of future time points. In this scenario of sequential anticipation, the brain must predict each moment, suppress what follows, and prepare for the subsequent crucial moment in a timely manner. However, this dynamic, contiguous process of temporal anticipation is not fully captured by previous studies, which primarily focus on situations where only the timing of a single target event is anticipated. Examples of these situations include anticipating the onset of an event, estimating the duration of an interval, or predicting the end of a lasting event (*Coull and Nobre, 1998*; *Martens and Johnson, 2005*; *Niemi and Näätänen, 1981*; *Nobre et al., 2007*; *Christina Nobre, 2001*; *Samaha et al., 2015*). In these studies of 'single' temporal anticipation, the task is considered complete once the target moment is reached, with no further process required. In this sense, sequential temporal anticipation offers a more thorough understanding of temporal anticipation compared to previous studies that only anticipate a single target event. This is the reason we have chosen to study sequential temporal anticipation.

We aim to investigate the brain's initiation of sequential temporal anticipation, as in the ecological case of music playing. One experimental model for investigating temporal anticipation involves presenting a series of sensory events, such as tones or visual images, and requiring human participants to predict the timing of these events (*Haegens and Zion Golumbic, 2018*; *Jones et al., 1981*; *Rimmele et al., 2018*; *Zanto et al., 2006*). However, sensory events offer external timing cues. When rhythmic sequences are presented, temporal anticipation can be based on neural oscillatory patterns that are entrained by these sequences that provide external timing cues (*Arnal et al., 2015*; *Doelling et al., 2019*; *Doelling and Poeppel, 2015*; *Henry et al., 2014*; *Henry and Obleser, 2012*; *Herrmann et al., 2016*; *Morillon et al., 2014*; *Morillon et al., 2016*; *Spaak et al., 2014*; *Ten Oever et al., 2017*; *van Bree et al., 2021*; *Tal et al., 2017*). Even when the intervals between events in a sequence are varied, the sequence can still serve as a temporal context for anticipating an incoming event, once the temporal contingencies of the sequence are learned (*Breska and Deouell, 2017*; *Breska and Ivry, 2020*; *Morillon et al., 2016*; *Paton and Buonomano, 2018*; *Rimmele et al., 2018*; *Wilsch et al., 2015*). For example, the timing of an upcoming event can be predicted by using the previous event as a temporal cue and estimating the duration of a fixed interval. Consequently, it becomes challenging to isolate the sequential temporal anticipation that is internally initiated by the brain – a new paradigm is required in this case.

We explored how the brain anticipates a series of prospective moments without the need for external timing cues. We adapted a straightforward tone-in-noise model, asking human participants to identify a faint tone at one of three set time points within a segment of white noise. To successfully detect the faint target tone, participants had to remember the time points and anticipate them in sequence within the constant white noise. This can be considered as memory-guided temporal anticipation (*Cravo et al., 2017*; *Grabenhorst et al., 2019*; *Mattiesing et al., 2017*; *Nobre and Stokes, 2019*; *Nobre and van Ede, 2018*), not for the timing of a single event, but for three potential moments. The white noise segment provides a global temporal context, limiting the time points of the target tone; the tone time points must be defined by referencing the onset of the noise or the duration of the noise period. Past research has indirectly studied temporal anticipation by observing its effects on behavioral and

neural responses to sensory events (*Grabenhorst et al., 2019*; *Tavano et al., 2019*) and by employing an encoding framework (*Herbst et al., 2018*). However, here, we directly investigate how the brain internally manages sequential temporal anticipation within flat white noise.

In addition to testing human participants, to gain insights on the functional aspects of our neural findings and uncover computational mechanisms underlying the sequential temporal anticipation, we turned to RNNs trained under the same behavioral paradigm in the current study. One advantage of employing neural network models is that RNNs are flexible to train and can be tested on variations of the current behavioral task to test different hypotheses (*Yang and Wang, 2020*). If the trained RNNs can indeed accomplish the behavioral task used in the current study and show a task performance comparable to the human participants' performance, we then study RNN hidden state activities and structures to gain insights on what procedures are conducted to accomplish the tone detection task and anticipate the temporal locations (*Yamins et al., 2014*; *Yang et al., 2019*).

It is challenging to identify internal anticipating processes without resorting to any external markers. We developed a data-driven approach, using modulation spectrum analyses on group-level neural power and phase measurements, to study internal anticipation processes. This method relies on the premise that if all human participants anticipate the same temporal structure, their neural dynamics should be similar. We found shared neural modulation components in the alpha-band range (12–30 Hz) that marked anticipated temporal locations and were correlated with tone detection performance. These could not be fully explained by expectation violation or false alarms. Next, we analyzed 12 RNNs trained on task variations to understand the functional role of the neural power modulation. This comparison revealed a mechanism of dynamic gain control in sequential temporal anticipation and suggested new hypotheses for future research.

## Results

Neural tagging of prospective temporal locations and behavioral results To aid participants in remembering prospective temporal locations, we presented a 2 s long white noise segment in each trial. A faint tone, calibrated to each participant's tone detection threshold (*Figure 1—figure supplement 1A*), was embedded at one of three time points: 500 ms, 1000 ms, and 1500 ms (*Figure 1A*). This consistent spacing was intended to assist in identifying neural correlates of sequential temporal anticipation within the white noise, without the need for external markers (for more details, see *Methods*). Once the noise segment ended, participants indicated if they had heard a tone. In 25 percent of the trials, only white noise segments were used ('no-tone' trials), while in the remaining 75 percent, the target tone was placed at one of the three temporal locations with equal probability ('tone' trials), (*Figure 1B*; *Figure 1—figure supplement 1B*). The sequence of tone and no-tone trials was randomized, leaving participants unsure of when or if the target tone would occur. As a result, participants had to remember the temporal locations within the noise segments and anticipate each one to effectively detect the faint tone. This task was carried out over three experimental blocks to accumulate an adequate number of 'no-tone' trials, which would help in capturing the neural signature of sequential temporal anticipation in flat noise.

We initially conducted conventional analyses on behavioral measurements and neural responses to the target tone. These analyses not only replicated previous findings but also confirmed that our task effectively manipulated the timing of the target event and modulated the sensory processing of the target tone (*Figure 1C*, *Figure 1—figure supplement 1C-E*; *Herbst et al., 2018*; *Ng et al., 2012*; *Nobre et al., 2007*; *Rimmele et al., 2011*; *Tavano et al., 2019*). We then shifted our focus to the no-tone trials. The target tone's three temporal locations were deliberately spaced 500 ms apart within the 2 s noise segments. If the brain anticipates the temporal locations on a sub-second scale, we should be able to detect neural signatures in power or phase measurements around 2 Hz (*Figure 1D*). Alternatively, the brain may solely encode the elapsed time of the noise segment to continually anticipate the target tone, without considering the exact temporal location. In this scenario, we wouldn't observe a 2 Hz component, but rather a neural signature corresponding to the length of the noise segment or the trial.

## Robust power modulations identified at 2 Hz and in the low frequency range

We sought to identify robust neural modulations for temporal anticipation in power and phase measurements using a data-driven method, a group-level modulation spectrum analysis (*Figure 1D*).

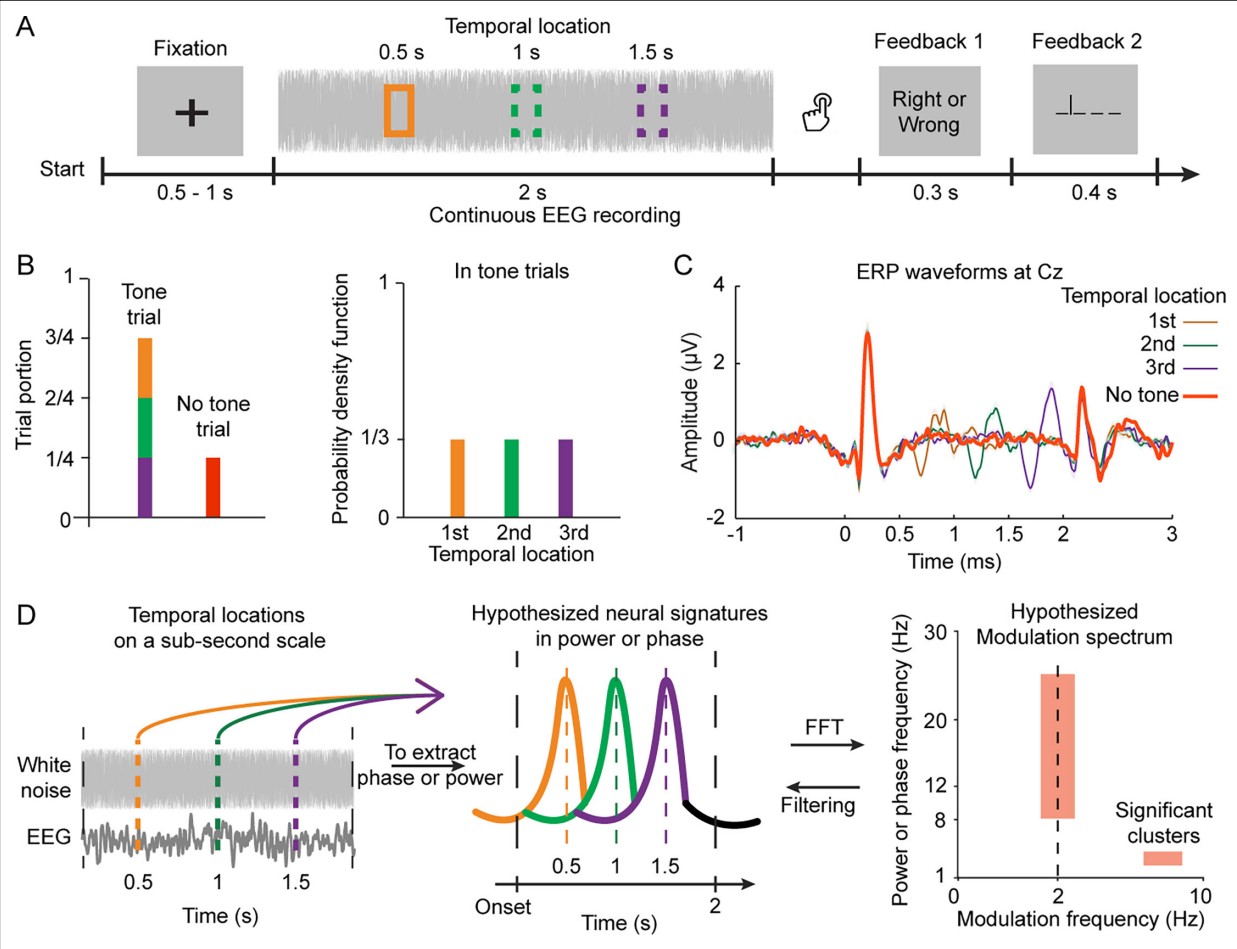

**Figure 1.** Experimental paradigm, target tone event-related potential (ERPs), and analysis procedures. (**A**) Tone Detection Task: Each trial presented participants with a 2 s white noise piece, after which they identified if a tone was heard. The target tone could appear at 500 ms, 1000 ms, or 1500 ms (indicated by colored boxes) or not at all. Two visual feedbacks were given: the first confirmed response accuracy, the second revealed the tone location. (**B**) Trial Portions and Target Tone Probability: The target tone was present in 75% of trials ('tone trials'), with the remaining trials only featuring white noise ('no-tone trials'). In tone trials, the target tone could appear at any of the three temporal locations, each with a 0.33 probability. (**C**) ERP Waveforms of the Cz Channel: Larger neural responses were evoked by the target tone at later temporal locations. (**D**) Hypotheses and Analysis Procedures for Sequential Temporal Anticipation: We hypothesized that participants would anticipate the temporal locations within the noise pieces to improve target tone detection. We expected to observe neural signatures reflecting anticipating processes for the temporal locations on a sub-second scale. To identify relevant neural components, we used a data-driven method, first performing a time-frequency decomposition on recorded EEG signals in no-tone trials to extract neural patterns in phase or power at each EEG frequency. We then conducted another fast Fourier transformation on the decomposed neural data to derive a modulation spectrum. After surrogate tests, we expected to find significant modulation clusters corresponding to the timescale of the temporal locations. We then filtered the EEG data according to each cluster's spectral range for further analysis.

The online version of this article includes the following figure supplement(s) for figure 1:

**Figure supplement 1.** Tone detection threshold, condition probabilities, and event-related potential (ERP) results.

We first combined all no-tone trials from the three blocks and, following time-frequency decomposition, obtained power and inter-trial phase coherence (ITPC). We then applied another Fourier transform to each frequency of the power and the ITPC to derive their modulation spectra. Subsequently, we carried out surrogate tests to pinpoint significant clusters in the modulation spectra. The power in this context was initially derived on each trial and then averaged over all trials; the ITPC primarily measures phase-locked, or event-evoked, neural responses (*Makeig et al., 2004*; *Mazaheri, 2022*). For a more detailed discussion on these two different measures, please refer to the *Methods*. It's also important to note that the modulation analysis used here is entirely distinct from the modulation power spectrum used in acoustic analysis (*Elliott and Theunissen, 2009*), which involves a 2D FFT directly applied to the spectrogram of sounds.

The modulation spectrum analyses indeed uncovered a rich profile of neural modulations in the power modulation spectrum (PMS) (*Figures 1C and 2A and B*). Although certain indications can be observed in the spectrograms derived from conventional power analyses (*Figure 2A*, *Figure 2—figure supplement 1A, B*), the modulation spectrum analysis directly singles out significant power frequencies and modulation frequencies, eliminating the need for post hoc examination. We only identified one cluster in the modulation spectrum of ITPC, which signals the onset and offset responses to the noise segments (*Figure 2—figure supplement 2*). As a result, we did not further investigate the neural phase domain or the phase-locked neural responses.

In the PMS, four significant modulation clusters were identified, as depicted in the right plot of *Figure 2B*. The time window used in the PMS calculation was 1.5 s, ranging from 0.25 s to 1.75 s after the onset of noise. Modulation components below 1.33 Hz, which did not construct two cycles over 1.5 s likely represented ramping neural activities. In contrast, modulation components above 1.33 Hz are considered rhythmic components. Therefore, the modulation frequency of 1.33 Hz acts as a demarcation line, separating ramping components from rhythmic ones, as seen in the left plot of *Figure 2B*, referred to as the Trend bound. As a result, the two clusters (Cluster 3 and 4) located above the Trend bound may represent the sequential temporal anticipation of local temporal locations. The topography of each cluster is displayed in *Figure 2C*. Subsequently, we analyzed the temporal dynamics of these four clusters.

The power dynamics for each cluster were derived by filtering the power according to the selected ranges of power frequency and modulation frequency, as shown in *Figure 2B* (refer to *Methods*). Cluster 1's power frequency range falls within the delta band (1–4 Hz). The temporal modulation trajectory of Cluster 1 mirrors the duration of the noise pieces, implying that the brain is actively monitoring the elapsed time of noise pieces and mentally registering the length of the noise piece. This aligns with previous research suggesting a correlation between neural signals in the delta band or slow ramping activities and interval (or event duration) estimation (*Breska and Deouell, 2017*; *van Wassenhove and Lecoutre, 2015*). This correspondence serves to validate our method used in this study. However, as this was not our primary focus, we did not further explore this cluster.

Clusters 2, 3, and 4 do not seem to correspond directly with temporal locations. However, we hypothesized that these clusters might collectively represent sequential temporal anticipation. To test this, we measured the power coupling between each pair of these clusters to understand their interrelation. We not only confirmed the previously found coupling relationship between the delta (Cluster 1) and the beta (Cluster 2) bands (*Arnal et al., 2015*; *Figure 2—figure supplement 1C*), but also discovered significant coupling relationships post-noise onset between Clusters 2 and 3, and Clusters 3 and 4. Furthermore, Clusters 2 and 4 shared the same power frequency range. The coupling relationships and power frequency overlaps suggested that Clusters 2, 3, and 4 might collectively represent a shared neural process. Consequently, we grouped them into a single entity, Cluster 234. As shown in *Figure 2E*, Cluster 1 and Cluster 234 concurrently represent the total elapsed time of the noise piece and the local temporal locations.

The reason Clusters 2, 3, and 4 are inherently related yet individually identified in the PMS is likely due to the Fourier transform in the PMS calculation, which requires components to be orthogonal. Other data-driven decomposition methods might be more effective in this case (*Kevric and Subasi, 2017*). However, the PMS supports an underlying psychological hypothesis specific to our study (that, as mentioned in the introduction, common neural dynamics should be shared among participants performing the same anticipation task. For more details, refer to *Methods*), and provides clear results. Therefore, we believe the PMS is suitable for our current purposes, though exploring other decomposition methods in future research could be beneficial.

We then turned our primary attention to Cluster 234, as it effectively captured the temporal anticipation of local temporal locations and represented a neural signature that, to the best of our knowledge, has not been previously identified. We removed the baseline of Cluster 234, a downward trend that spanned the entire trial (for more details, see *Methods*), and plotted the dynamics of the corrected Cluster 234 in *Figure 2F*. We also depicted the topography of its modulation strength from 0.4 s to 1.9 s in *Figure 2H* (refer to *Methods* for further details). It is evident that the neural peaks in the corrected Cluster 234 denote the three temporal locations in the noise pieces (*Figure 2F*). After fitting a Gaussian curve to each peak, we found the peak latencies, relative to the corresponding temporal locations, to be 0.24 s, 0.16 s, and 0.13 s, respectively (*Figure 2G*). This suggests a precession of

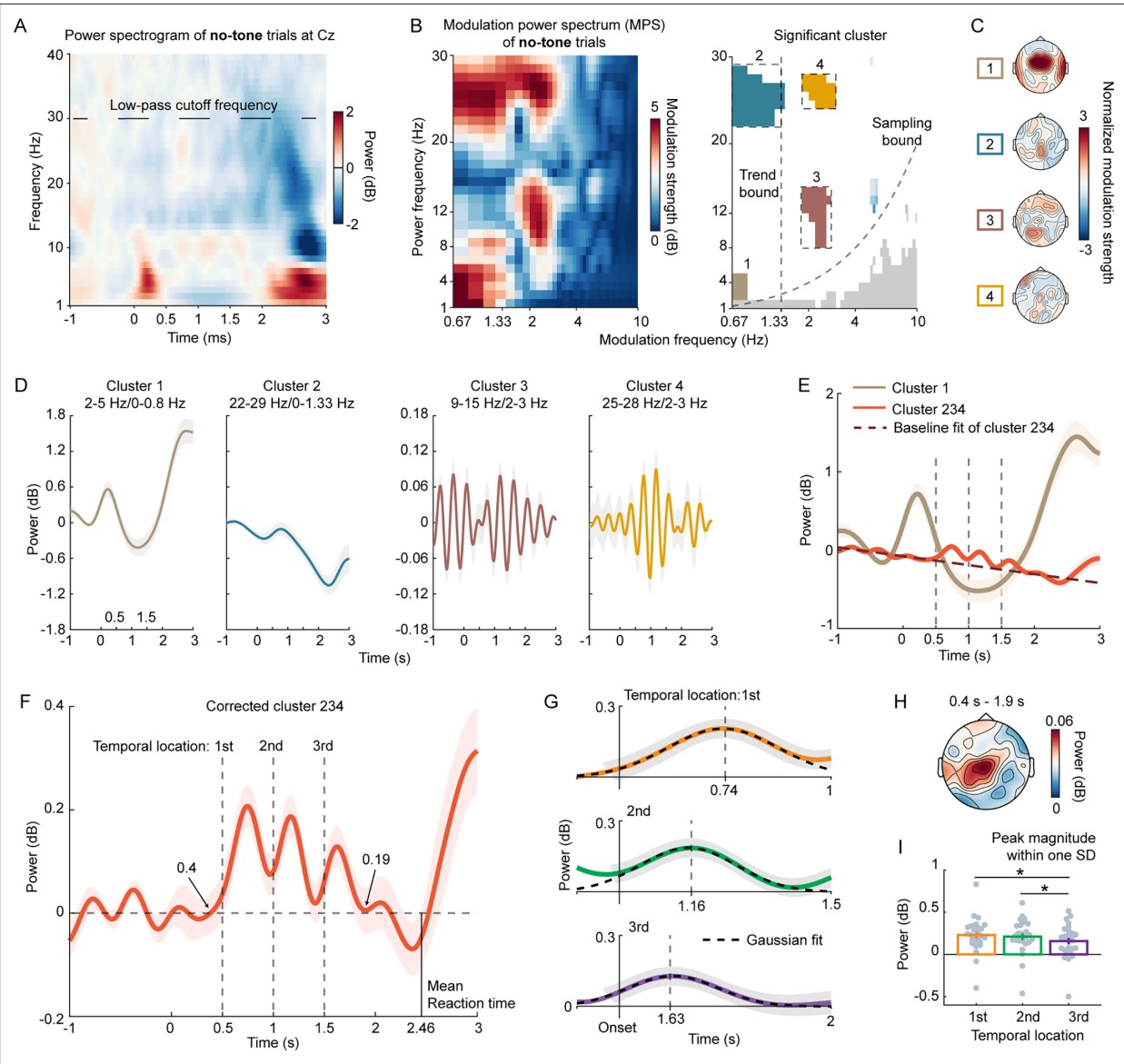

**Figure 2.** Exploring power modulation spectrum (PMS) and power dynamics. (**A**) Displays a spectrogram of induced power from all no-tone trials. (**B**) Presents raw PMS (left plot) and significant clusters (right plot). The analysis was conducted on the EEG channel of Cz as previous research shows that anticipation-related neural signals can be largely captured by the Cz (see *Methods*). The y-axis signifies the frequency of induced power, or power frequency, while the x-axis represents the modulation frequency. In the right plot, colored bins indicate significant modulation bins with high modulation strength, as determined by a surrogate test. The four dashed boxes highlight the selected power and modulation frequency ranges for the clusters. Modulation frequencies larger than the Trend bound denote rhythmic neural components, while those smaller than the Trend bound represent ramping neural components. Modulation bins below the Sampling bound violate the Nyquist theorem and are, therefore, not meaningful. (**C**) Topographies of significant modulation clusters for visualization, with green dots indicating significant EEG channels. (**D**) Dynamics of significant clusters, where the induced power was filtered according to the clusters selected in (**B**), showing the power dynamics of each cluster. (**E**) Merged clusters. Based on the power-power coupling between each pair of clusters (*Figure 2—figure supplement 1*) and the power frequency overlaps, we grouped Clusters 2, 3, and 4 into one - Cluster 234. Cluster 1 and Cluster 234 collectively capture both the local temporal locations and the global elapsed time of the noise piece. (**F**) Corrected cluster 234, derived by removing the baseline of Cluster 234 from −1 s to 0 s. (**G**) Gaussian fits to each neural peak of Corrected cluster 234, providing peak latencies (mean of Gaussian fits) and peak widths (standard deviation). (**H**) Topography of modulation strength of Corrected cluster 234 from 0.4 s to 1.9 s post noise onset. (**I**) Peak magnitude within one standard deviation (SD) of the Gaussian fits. Each gray dot represents individual participant data. Shaded areas of color and error bars signify ±1 standard error of the mean across participants (n=27). * denotes $p<0.05$.

The online version of this article includes the following figure supplement(s) for figure 2:

**Figure supplement 1.** Power coupling analysis.

**Figure supplement 2.** Modulation spectrum analysis of inter-trial phase coherence (ITPC).

neural dynamics along the temporal locations. We then calculated the average peak magnitudes for each temporal location within one standard deviation (SD) of the corresponding Gaussian fit and conducted a one-way rmANOVA. A significant main effect for Temporal location was observed ($F(2,52) = 4.30$, p=0.019, $\eta_p^2$p20.142). The peak magnitudes of the first and second temporal locations were significantly larger than that of the third temporal location (*Figure 2I*) (paired t-test: $t(26) = 3.38$, p=0.038; $t(26) = 2.41$, p=0.045). To control the false positive rate of multiple comparisons, we applied an adjusted False Discovery Rate (FDR) correction throughout the entire paper (*Benjamini and Hochberg, 1995*; *Yekutieli and Benjamini, 1999*).

One might reasonably assume that the corrected Cluster 234 in the no-tone trials reflects an expectation violation. This is because participants may anticipate a target tone at a temporal location, and its absence could generate a prediction error (*Friston, 2005*) or a mismatch (*Heilbron and Chait, 2018*; *Näätänen et al., 2007*), potentially inducing the neural peaks of the corrected Cluster 234. If this were the case, the magnitudes of these neural peaks should increase from the first to the third temporal location, as the likelihood of the target tone's appearance increases with the elapsed time of the noise piece (*Janssen and Shadlen, 2005*; *Nobre et al., 2007*; *Figure 1—figure supplement 1B*). However, our data show that the peak magnitude of the corrected Cluster 234 actually decreases with the temporal location (*Figure 2I*), making the expectation violation explanation unlikely. This decreasing trend in peak magnitude may correlate with the estimation of the likelihood of a trial being a tone trial (*Grabenhorst et al., 2021*; *Figure 1—figure supplement 1B*), but our subsequent analyses fitting this likelihood to individual participants' data did not reveal a significant effect (p>0.05). Further studies are required to specifically test this hypothesis.

We analyzed the power dynamics in the conventionally defined frequency bands and observed indications of power marking the temporal locations in the no-tone trials in both the beta2 band (21–30 Hz) and the alpha band (8–12 Hz). This was after we intentionally divided the beta band (12–30 Hz) into the beta1 (13–20 Hz) and beta2 bands (21–30 Hz) (*Figure 3—figure supplement 1A*). Our data-driven method here aligns with standard analysis procedures, but it uniquely extracts key neural features from EEG signals. This is achieved without initially plotting power dynamics in predefined frequency bands for observation and then conducting post-hoc analyses.

## Correlation between behavioral measurements and corrected Cluster 234

In the preceding analyses, we consolidated all no-tone trials from the three experimental blocks to ensure adequate statistical power for the PMS analysis. This was aimed at identifying the neural signature associated with sequential temporal anticipation, which we were uncertain could be detected a prior. After establishing Corrected Cluster 234, we proceeded to examine each block individually. This allowed us to explore the correlation between behavioral performance and Corrected Cluster 234, as well as to assess any potential learning effects across the blocks.

*Figure 3A* illustrates Corrected Cluster 234 in each block, showing that both the peak magnitudes and peak latencies varied according to the blocks and temporal locations. By following the procedure of fitting Gaussian curves to the power peaks (as shown in *Figure 2G*), we derived the peak magnitudes in each block (*Figure 3B*, *Figure 3—figure supplement 1*). We then conducted a two-way rmANOVA, with Temporal location and Block number as the primary factors. The main effect of Temporal location was significant ($F(2,52) = 5.01$, p=0.010, $\eta_p^2$p20.162), whereas the main effect of Block number ($F(2,52) = 0.51$, p=.604, $\eta_p^2$p20.019) and the interaction ($F(4,104) = .25$, p=0.909, $\eta_p^2$p20.010) were not. The peak magnitude displayed a decrease with the temporal location, as evidenced by a significant linear trend ($F(1,26) = 9.22$, p=0.005, $\eta_p^2$p20.262). The peak magnitude around the first temporal location was significantly larger than the third location ($t(25) = 3.04$, p=0.016; adjusted FDR correction was applied). These findings indicate that the block number did not interact with the neural coding of the local temporal locations, suggesting that a learning effect was not observed across blocks.

Contrary to our initial hypothesis that learning the timing of prospective moments would be a gradual process, participants quickly grasped the temporal locations of the target tone within the noise pieces. They were able to anticipate these locations even in the initial block. This may be attributed to the 10–15 trials in the training session we conducted prior to the formal test, which helped familiarize participants with the task. It's possible that participants learned the temporal locations during this training phase. Additionally, the visual feedback provided on the temporal location of the target

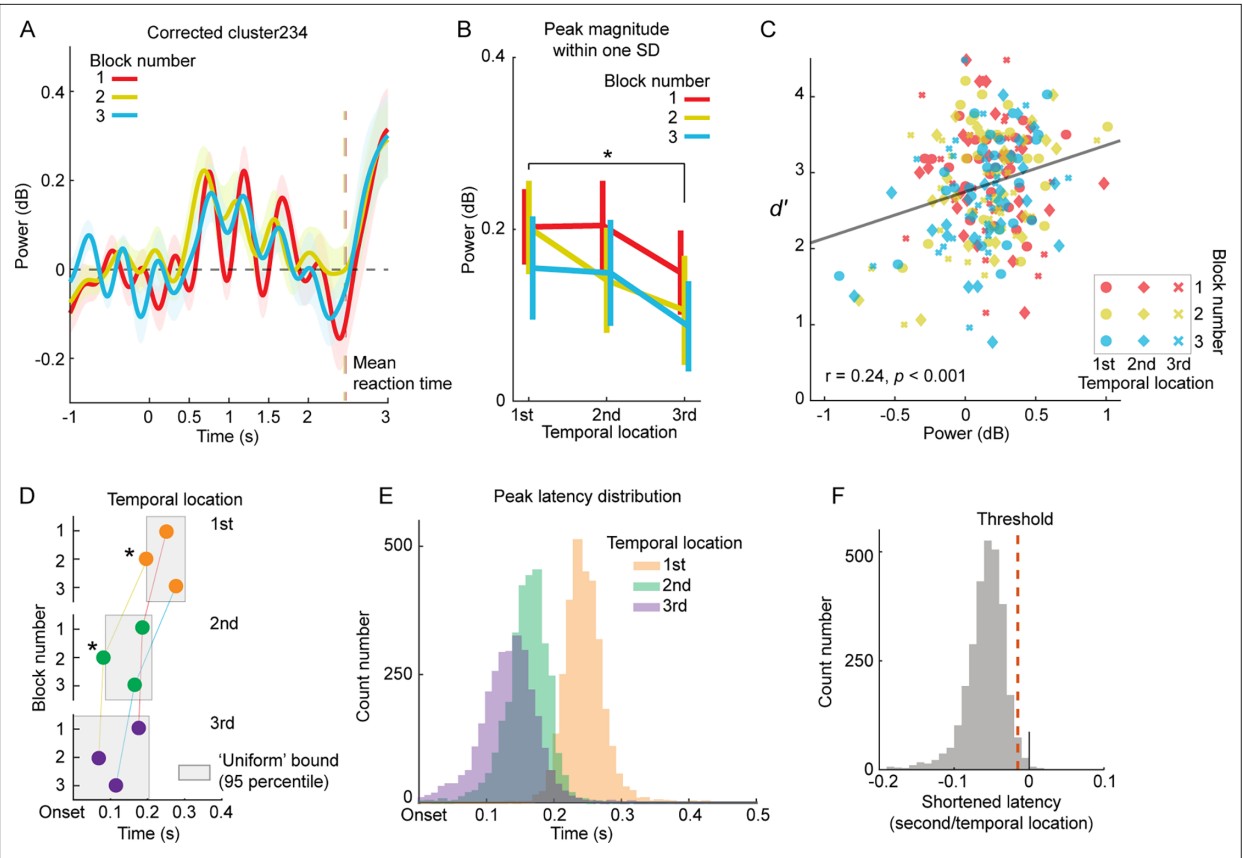

**Figure 3.** Analyzing power modulation correlation with tone detection performance and neural power precession. (**A**) Depicts power dynamics of Corrected cluster 234 in each block, with the colored vertical line representing the group mean of reaction time per block. (**B**) Illustrates peak magnitudes within one standard deviation (SD) of the Gaussian fits. A significant main effect of Temporal location was observed (p<0.05), but not for Block number (p>0.05). (**C**) Shows the correlation between d-prime values and neural peak magnitudes. Peak magnitudes of Corrected cluster 234, derived from no-tone trials, positively correlate with tone detection performance. Each dot represents individual participant data. (**D**) Presents peak latency differences of Corrected cluster 234 between blocks. Filled circles denote peak latencies derived from fitting Gaussian curves to Corrected cluster 234, with different temporal locations coded by the color scheme. Gray square areas indicate the 95% confidence interval of peak latencies from permutation tests across blocks, representing a 'Uniform' bound, a null hypothesis that all peak latencies at a temporal location do not significantly differ across blocks. If a peak latency falls outside the 'Uniform' bound, it significantly differs from other peak latencies. (**E**) Displays peak latency distribution. Three distributions for three temporal locations were derived from permutation tests. (**F**) Depicts distribution of shortened latency per temporal location. A set of peak latencies was randomly selected from distributions in (**E**) and a line was fitted to the selected peak latencies with the temporal location (1, 2, and 3) as the independent variable. This generated a distribution of slopes representing shortened latency per temporal location. 97.5% of the slope values fall below 0, suggesting a significant decrease in peak latency with the temporal location. The shaded areas of color in (**A**) and the error bars in (**B**) represent ±1 standard error of the mean across participants (n=27). * denotes p<0.05.

The online version of this article includes the following figure supplement(s) for figure 3:

**Figure supplement 1.** Analyses of Gaussian Fits to neural peaks Of corrected cluster 234 and their correlation with behavioral measurements.

tone during tone trials likely facilitated this swift learning process, as it offered clear information about each temporal location. Future studies could delve deeper into this learning process by eliminating the visual feedback. However, this efficient learning process proved beneficial in identifying the neural signatures of sequential temporal anticipation, which was our primary focus.

A lingering question is whether the neural power modulation, which marks the local temporal locations in the no-tone trials, is relevant to the participants' tone detection performance in the tone trials. By correlating the peak magnitudes of Corrected Cluster 234 with the d-prime values (*Figure 1— figure supplement 1C*, *Figure 3C*), we discovered a significant positive correlation (*r*=0.24, p<0.001). This indeed suggests that anticipation of the local temporal locations enhances tone detection. We further illustrate in *Figure 3—figure supplement 1B* the correlation between d-prime values and peak magnitudes in each block, and in *Figure 3—figure supplement 1C* the correlation at each temporal

location. One possible explanation for the emergence of Corrected Cluster 234 is that each neural peak following each temporal location is due to false alarms - participants may erroneously perceive a tone within the white noise, thereby inducing a neural peak after a temporal location. If this were the case, the false alarm rate should positively correlate with the peak magnitudes of Corrected Cluster 234 in the no-tone trials. However, we found that, for the most part, the false alarm rates negatively correlate with the peak magnitudes (*Figure 3—figure supplement 1D*) . Therefore, Corrected Cluster 234 was not a result of false alarms in the no-tone trials.

## Precession of power dynamics suggests active anticipation

We next sought to quantify the observation from *Figure 3A* that the peak latencies of Corrected Cluster 234 varied according to the block number and temporal location. We performed statistical tests on the peak latencies at the group level using permutation tests (refer to *Methods*), as the peak latencies could not be reliably derived using the Gaussian fits procedure on each participant's data. The permutation tests consisted of two steps. First, we examined whether the peak latencies changed across blocks. The permutation tests yielded a 95% confidence interval for each temporal location across the three blocks (refer to *Methods* for details). We discovered that the peak latencies at the first and second temporal locations of the second block were significantly shortened (*Figure 3D*). Second, we investigated whether the peak latencies changed across temporal locations. We combined the peak latency samples generated from the permutation tests across blocks to form a peak latency distribution for each temporal location (*Figure 3E*). We then fitted a line to three peak latencies randomly selected from the three distributions. The distribution of the slopes of the fitted lines indicates that the peak latency significantly decreased with the temporal location (*Figure 3F*). This observation is likely due to the sequential anticipation of the temporal locations - the anticipation of the first temporal location facilitated the anticipation of the second, and so on. We further examined this observation in the subsequent RNN perturbation analyses.

## Task-optimized RNNs replicate the behavioral and neural findings and suggest a process of dynamic gain control

We have discovered a unique neural signature, labeled as Corrected Cluster 234, which identifies local temporal locations within the 2 s white noise segments (*Figures 2 and 3*). This neural signature cannot be explained by either expectation violation (*Figure 1—figure supplement 1B*) or the false alarm rate (*Figure 3—figure supplement 1D*). The question then arises: what functional role does this neural signature play in the task at hand? To answer this, we trained continuous-time Recurrent Neural Networks (CRNNs) to perform the same task. The logic behind this is that the CRNNs, when trained to perform the same task as human participants, might also perform certain computations in a manner similar to the human brain (*Yamins and DiCarlo, 2016*; *Yamins et al., 2014*). If the trained CRNNs demonstrate a behavioral performance similar to human participants, we will study the internal dynamics of the trained RNNs to understand how they accomplish the task. Furthermore, the flexibility of RNNs allows us to test various conditions related to the current study, which can help generate new hypotheses for future research.

We developed a CRNN structure adhering to Dale's rule (*Dale, 1935*; *Song et al., 2016*) and created inputs for the CRNNs that encapsulated the essence of the behavioral task (*Figures 1A and 4A*). Detailed information about the CRNN parameters and training procedures can be found in the *SI Appendix, under Methods*. We systematically adjusted the external noise level of the inputs while maintaining a constant target tone magnitude. This is because if the timing of temporal locations was expected to assist in detecting the weak target tone, the sequential temporal anticipation should be deployed more extensively at high external noise levels (low Signal-to-Noise Ratio - SNR) and less so at lower noise levels (high SNR). As a result, the hidden activities of the CRNNs trained at high noise levels should exhibit dynamics with 2 Hz modulation components indicating the temporal locations, but this should be absent or reduced at lower noise levels. If this is the case, it implies that the 2 Hz neural component depicted in *Figures 2 and 3* is associated with the brain's use of sequential temporal anticipation to deal with external sensory noise.

We trained five separate CRNNs at five different noise levels, specifically: 0.01, 0.05, 0.1, 0.15, and 0.2 standard deviations (SD) of added Gaussian noise. We then calculated the d-prime values for each trained CRNN at various testing noise levels (*Figure 4B*). For each of these five CRNNs, we chose

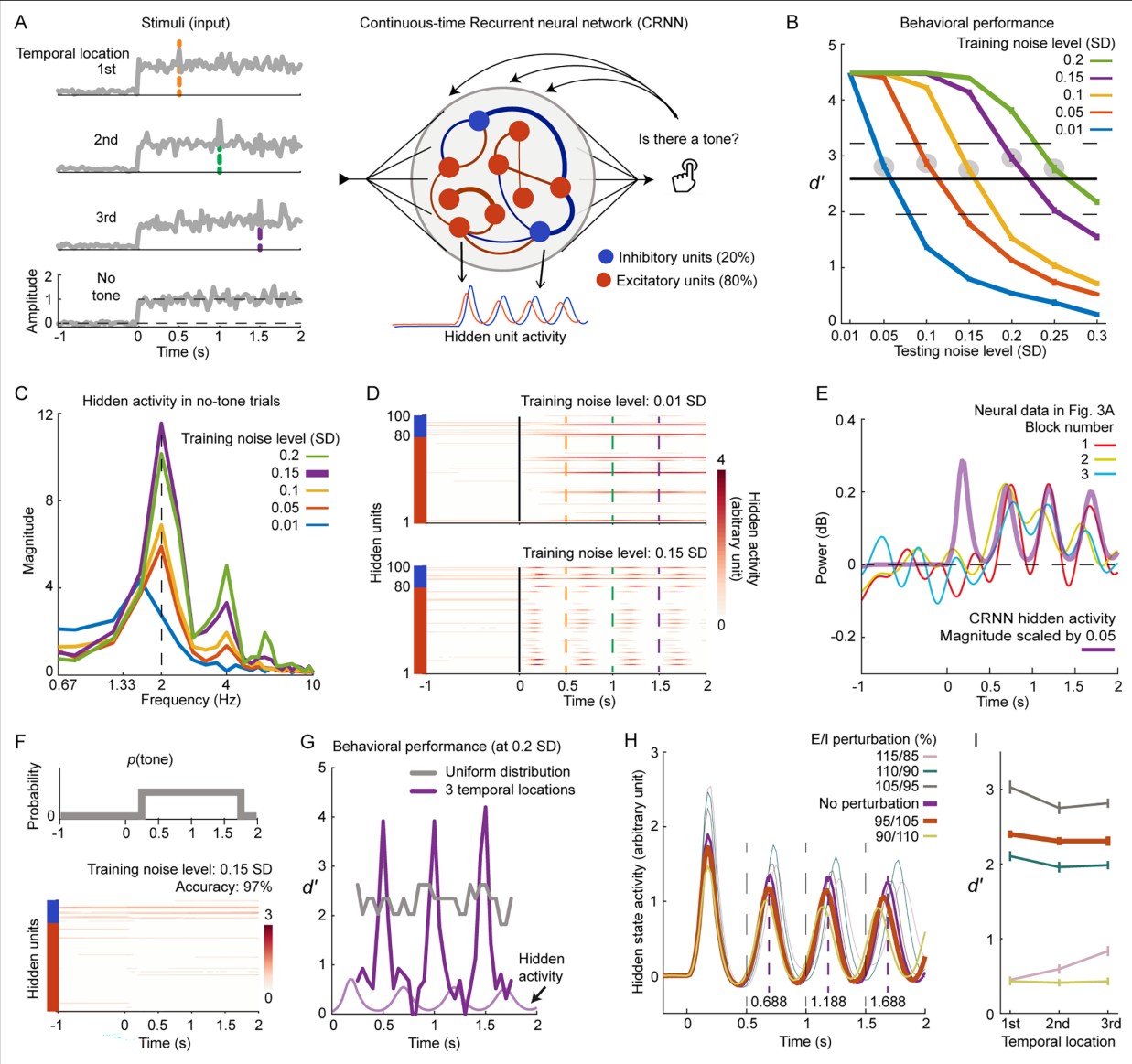

**Figure 4.** Behavioral and neural findings explained by task-optimized continuous-time recurrent neural networks (CRNN). (**A**) Training procedures for CRNNs are depicted. The left plot shows the inputs to CRNNs, while the right plot outlines the CRNN structure and training procedures. (**B**) Task performance of trained CRNNs is presented. Five CRNNs were trained at five different external noise levels and tested at seven noise levels. The solid black line represents the group mean of human behavioral performance (*Figure 1—figure supplement 1*), with dashed black lines indicating ±2 standard errors across participants. Filled gray circles mark the testing noise levels chosen for the corresponding CRNNs whose performance matches human performance. (**C**) Modulation spectra of hidden activities of the CRNNs at the corresponding testing noise level selected in (**B**) are displayed. Modulation spectra of CRNN hidden activities were derived following the EEG analysis procedures. (**D**) Hidden activities of two CRNNs derived from 40 no-tone trials are shown. The y-axis specifies whether the units are excitatory (red units) or inhibitory (blue units). (**E**) Compares CRNN hidden activities with neural data. The neural data are sourced from the group-mean data shown in *Figure 3A*. The CRNN analyzed was trained at the noise level of 0.15 SD and tested at the noise level of 0.2 SD. (**F**) Training CRNN with a uniform probabilistic distribution of tone locations is presented. The upper plot illustrates a trial structure with a uniform distribution of tone locations, where a tone can appear at any time from 0.25 s to 1.75 s post noise onset with equal probability. The lower plot reveals that the hidden activities of the CRNN trained using the 'uniform' trial structure did not exhibit any rhythmic activities. (**G**) Task performance in the 'uniform' trial structure is shown. At the testing noise level of 0.2 SD, the CRNN trained in (**F**) and the CRNN trained with three temporal locations at the training noise level of 0.15 SD were tested. (**H**) Hidden activities of CRNNs with perturbed excitation-inhibition (EI) weights are depicted. The thick colored lines refer to the EI perturbations that caused the peak latencies of CRNN hidden activities to decrease with the temporal location. The violet line represents the unperturbed CRNN selected from (**C**), with dashed vertical lines denoting peak latencies of this unperturbed CRNN. (**I**) Task performance of perturbed CRNNs at the testing noise level of 0.2 SD is presented. Shaded areas of color and error bars signify ±1 standard error of the mean across participants (n=27).

a specific testing noise level at which the corresponding CRNN produced a task performance that matched the average human behavior performance (*Figures 1C and 4B*). Following this, we analyzed the activities of the hidden units within the selected CRNNs and calculated the modulation spectra of their hidden activities during the presentation of 40 no-tone trials. This was done by using the same procedure as the one used to quantify power modulations of neural signals (*Figure 4C*). As expected, the modulation strength around 2 Hz increased with the external noise level used in training the CRNNs. The CRNN trained at the noise level of 0.15 SD exhibited the highest modulation strength at 2 Hz, while the CRNN trained at 0.01 SD showed the least modulation strength (*Figure 4C*). This is also reflected in the temporal dynamics of their hidden activities (*Figure 4D*). We averaged the hidden activities of the CRNN trained at 0.15 SD across all hidden units, similar to EEG recording where neural signals from both excitatory and inhibitory synapses are combined, and plotted the averaged hidden activities of the CRNN alongside Corrected Cluster 234 (*Figure 4E*). The results showed that the dynamics of the CRNN hidden activity closely mirrored the power dynamics of Corrected Cluster 234.

One potential explanation is that the 2 Hz modulation components observed in both the brain and the CRNNs performed a noise suppression operation – the noise period outside of the tone locations was suppressed to facilitate the detection of the target tone. If this is the case, a target tone might be suppressed along with the noise if it occurs at a time point other than the three temporal locations used for training the CRNNs. Furthermore, could a CRNN display such 2 Hz hidden dynamics simply due to high noise levels, and not because of the specific temporal locations used? To answer these questions, we trained a CRNN with a new trial structure. In this structure, the training noise level was set at 0.15 SD and the target tone could appear at any time between 0.25 s and 1.75 s after the noise onset (*Figure 4F*, upper panel) - a 'uniform' trial structure. The newly trained CRNN exhibited flat hidden activities devoid of any rhythmic patterns (*Figure 4F*, lower panel). Subsequently, in the uniform trial structure at a noise level of 0.2 SD, we tested the newly trained CRNN and the CRNN trained with the three fixed temporal locations (both trained at a noise level of 0.15 SD). The tone detection performance of the newly trained CRNN dropped significantly but remained within a moderate range; in contrast, the CRNN trained with the three fixed temporal locations demonstrated superior tone detection performance at the temporal locations but much poorer performance at other time points (*Figure 4G*). It should be noted that the calculation of d-prime values here differed from *Figure 4B* as different trial structures were used (refer to *Methods* for details).

The findings presented in *Figure 4F&G* suggest that the 2 Hz neural modulation observed in Corrected Cluster 234 served to enhance sensory sensitivity to the target tone at the anticipated temporal locations, while selectively suppressing sensory noise during irrelevant noise periods. Interestingly, the power frequencies of Corrected Cluster 234 fall within the range typically defined as the alpha and beta bands (*Figure 2B*). Increasing evidence has shown that neural activities within these alpha-beta bands are closely associated with the suppression of internal/external noise and the regulation of information flow (*Arnal and Giraud, 2012*; *Haegens et al., 2012*; *Jensen and Mazaheri, 2010*; *Spitzer and Haegens, 2017*; *Wöstmann et al., 2019*; *Wöstmann et al., 2017*; *Zumer et al., 2014*).

Our neural findings, in conjunction with the CRNN analyses, indicate that the process of sequential temporal anticipation was partially executed as a gain control mechanism over time (*Salinas and Sejnowski, 2001*; *Strauß et al., 2014*; *van Ede et al., 2018*). This is similar to how the brain selectively filters information for future time points, akin to the way it processes sensory information in space by focusing on attended locations and disregarding irrelevant background noise (*Hillyard et al., 1998*; *Reynolds and Chelazzi, 2004*).

## Enhanced inhibitory activity explains precession of power dynamics

Previous studies have demonstrated that disrupting the balance between excitation and inhibition alters the pace of neural oscillatory activities (*Brunel and Wang, 2003*). In our study, we hypothesized that if the temporal locations were constant, the expedited neural peak latencies of Corrected cluster 234 (*Figure 3E&F*) could have been stimulated by an increase in inhibitory or excitatory neural activities during the anticipation process. Following this, we disrupted the excitation-inhibition balance in the CRNNs to investigate how alterations in this balance influenced the peak latencies of hidden CRNN activities.

In the CRNN trained at a noise level of 0.15 SD, we manipulated the connectivity strength of excitatory and inhibitory units in such a way that the original ratio between excitatory and inhibitory projection weights (E/I ratio) was systematically adjusted (*Figure 4H*). For instance, we reduced the weights of excitatory units by 5% and correspondingly increased the weights of inhibitory units in a CRNN by 5%. This resulted in weaker projections from excitatory units to inhibitory units, and stronger projections from inhibitory units to excitatory units. We then calculated the average hidden activities of the CRNNs with different E/I ratios (*Figure 4H*). When the E/I ratio fell below one (e.g. 95/105) – indicating stronger inhibitory projections and weaker excitatory ones – the peak latencies decreased with the temporal location, mirroring the neural finding shown in *Figure 3E&F*. The peak latencies of the original CRNN (E/I ratio: 100/100) remained constant with the temporal location and were fixed at 0.188 s (the middle time point between the two largest points of each peak, indicated by the vertical dashed lines in *Figure 4H*). We then evaluated the behavioral performance of the CRNNs with altered E/I ratios and discovered that, when the E/I ratio of the CRNN was 95/105, the behavioral performance still fell within the range of human behavioral performance (*Figure 4I*).

In conclusion, our examination of E/I ratios in CRNNs suggests that the observed neural power precession in Corrected cluster 234 may be linked to enhanced neural activities of inhibitory circuits. The anticipations of earlier temporal locations facilitated the anticipations of subsequent ones, which is associated with the increased efficiency of inhibitory circuits. This aligns with previous neurophysiological findings and theories suggesting that neural signals in the alpha-beta bands correlate with inhibitory activities (*Klimesch et al., 2007*; *Lakatos et al., 2016*; *Sherman et al., 2016*).

## Discussion

To investigate how the brain sequentially anticipates future moments, we developed a 'preremembering' task by adapting a basic tone-detection paradigm, applied innovative modulation spectrum analyses to the electrophysiological data, and trained and analyzed 12 RNN models. Our research revealed that a neural modulation component in the alpha-beta band range was utilized to locally anticipate prospective temporal locations on a sub-second scale (*Figure 2*). The neural modulation markers for sequential temporal anticipation were found to positively correlate with the performance of tone detection (*Figure 3*), and these correlations could not be explained by expectation violation or false alarms. Utilizing task-optimized RNNs (*Figure 4*), we delved deeper into the functional role of our neural findings, demonstrating that the neural power modulations underpinned a gain control process over time that reduced irrelevant noise periods and heightened sensory sensitivity at the anticipated temporal locations. Collectively, our behavioral, neural, and neural network modeling findings suggest that temporal anticipation, a psychological term for a general task requirement, was executed in this context as a dynamic gain control process.

A key focus in psychology and neuroscience is understanding how the brain copes with noise and extracts signals (*Faisal et al., 2008*). The brain is constantly processing sensory inputs, yet relevant and useful information is often sparse in the real world. In the context of our study, meaningful events (i.e. the target tone) potentially occur at only one of a few time points, while sensory noise is a constant presence. Therefore, the functional requirement for temporal anticipation is not only to foresee prospective moments but also to disregard irrelevant noise periods. As shown in our neural and modeling findings, neural operations in the alpha and beta bands enhanced sensory sensitivity to the anticipated temporal locations and suppressed noise outside of these locations. This discovery aligns well with the function of attention in space – in studies of spatial attention, the alpha and beta neural bands have been found to correlate with the suppression of irrelevant objects or backgrounds (*Bauer et al., 2012*). Our findings suggest a reinterpretation of temporal anticipation, which has often been treated as a unitary construct in various contexts (*Nobre and van Ede, 2018*) – the precise underlying mechanisms of temporal anticipation can vary according to functional demands.

An alternate interpretation of Corrected Cluster 234 marking the temporal locations of the target tone (as seen in *Figures 2 and 3*) could be that this neural signature signifies the probability distribution of an event occurring over time (*Grabenhorst et al., 2019*; *Janssen and Shadlen, 2005*; *Figure 1B*). It would be beneficial to further investigate whether Corrected Cluster 234 aligns with the neural representations of probability distributions observed in animal studies by altering the distribution shapes (*Janssen and Shadlen, 2005*). However, it is not always required to represent a probability distribution of event occurrence, particularly when the signal-to-noise ratio (SNR) of tone to noise is

high, simplifying the tone detection task. Even when the probability distribution of tone occurrence remained constant, the RNN modelling results indicated a weakening of the neural dynamics marking the tone locations as the SNR decreased (i.e. the task became easier) (see *Figure 4C*). Therefore, it's plausible that the need to represent probabilities of event occurrences is influenced by task demands. Further research could certainly test this hypothesis, which is suggested by our RNN models.

Regarding the brain regions involved in sequential temporal anticipation, research has shown that, on a supra-second scale, the anticipation of incoming event timings and interval estimation are primarily processed in the parietal and motor areas (*Bolger et al., 2014*; *Coull et al., 2016*; *Janssen and Shadlen, 2005*; *Merchant and Yarrow, 2016*; *Morillon and Baillet, 2017*; *Wittmann, 2013*). In terms of local temporal location anticipation, significant modulation strength was mostly observed in the EEG channels around Cz (*Figure 2H*), leading us to hypothesize that the corresponding neural modulation component may originate from subcortical areas. One potential area is the basal ganglia, particularly the striatum, which has been linked to motor sequence generation and various sub-second scale timing tasks (*McClure et al., 2003*; *Mello et al., 2015*; *Motanis and Buonomano, 2015*; *Nozaradan et al., 2017*). As sequential actions necessitate coordination in a series of moments (*Rimmele et al., 2018*; *Schubotz, 2007*), this functional requirement aligns with the cognitive process required in our task, which is to anticipate a sequence of prospective moments.

In conclusion, we have identified a neural processing scheme involved in sequential temporal anticipation. The brain utilizes neural power modulations across several neural bands, specifically alpha and beta, to anticipate sequential local temporal structures. The task-optimized artificial neural network models, particularly the RNNs, have helped uncover a dynamic gain control process that underpins temporal anticipation. This process involves not only anticipating a few significant future moments, but also actively disregarding irrelevant events that frequently occur.

## Methods

### Participants

Thirty participants took part in the experiment (age 18–35, right-handed, 16 females). All participants had normal hearing and no neurological deficits according to their self-report. Written informed consent was obtained from each participant before the experiment; monetary compensation was provided accordingly after the experiment was finished or terminated. Two participants could not undergo the EEG recording because of the metal fillings or decorations above their neck and chose to terminate the experiment. One participant's data was excluded because triggers for stimulus onsets were not recorded during EEG recording. Therefore, the neurophysiological data shown in the main text included twenty-seven participants (age 18–35, right-handed, 14 females). The experimental protocol was approved by the Ethics Council of the Max Planck Society.

### Experimental paradigm and stimuli

The participants were instructed to detect a 1000 Hz pure tone of 30 ms embedded in a 2 s piece of white noise at 1 of 3 temporal locations: 500 ms, 1000 ms, and 1500 ms. In each trial, there was no cue to direct participants' attention to one of the three temporal locations. We hypothesize that the participants, not knowing when the target tone would appear, had to anticipate the three temporal locations of the target tone sequentially. Our current study focused on how the participants memorized the temporal locations and anticipated the temporal locations sequentially.

The experiment was comprised of three blocks and the experimental procedures are illustrated in *Figure 1A*. In each trial of each block, the participants were required to keep eyes open and to fix on a white cross in the center of a black screen. The white cross stayed on the screen for a duration randomly chosen between 1 s and 1.5 s. After the white cross disappeared, the 2 s noise piece was presented. The participants pressed one of two buttons to indicate whether they heard a tone after the noise piece was over. Visual feedback after participants' responses was shown on the screen for 0.3 s to inform the participants whether their response was 'correct' or 'wrong.' Then, another feedback was shown on the screen for 0.4 s to indicate whether and when the target tone appeared in the noise piece with one of four symbols: '_ _ _ _' (no tone was presented), '_ | _ _ _' (a tone appeared at 500 ms after the noise onset), '_ _ | _ _' (1000 ms), '_ _ _ | _' (1500 ms). After the two visual feedbacks,

the next trial started immediately. The feedbacks were designed to help the participants quickly learn the tone-detection task and memorize the temporal locations.

In each of three blocks, there were four conditions: a tone appeared at either 500 ms, 1000 ms, or 1500 ms, or no tone was presented. There were 40 trials for each condition in each block, so there were 160 trials in each of three blocks. 480 trials in total were presented. The conditions were randomly chosen in each trial and the trial order was randomized across blocks between participants. After each block, the participants could choose to take a break of 1 or 2 min or to start the next block immediately.

The white noise piece in each trial was independently generated using a random number generation function, 'rand,' in MATLAB R2016b (The MathWorks, Natick, MA) at a sampling rate of 20,000 Hz. As a new white noise piece was generated in each trial, the participants could not memorize the acoustic details of white noise to help them detect the target tone. The amplitude of white noise pieces was normalized to 70 dB SPL (sound pressure level) by referring the noise pieces to a 1-min white noise piece that was measured beforehand to be 70 dB SPL at the experimental setting. We applied a cosine ramp-up function in a window of 50 ms at the onset of the noise pieces and a sine ramp-down function of 50 ms at the offset.

The 1000 Hz target tone of 30 ms duration was generated using the function of 'sin' in MATLAB with a fixed initial phase of 0 degree. We applied a cosine ramp-up function in a window of 10 ms at the onset of the tone and a sine ramp-down function of 10 ms at the offset. As the threshold for detecting a pure tone in white noise varies across individuals, we set an individualized signal-to-noise ratio (SNR) of the target tone to the white noise for each participant, so that the difficulty level of the task could be comparable between participants and be controlled. We measured each participant's tone detection threshold before the formal experiment (Fig. S1A) and set the SNR of the target tone to be 3 dB above each participant's threshold.

To measure the tone detection threshold, we employed a two-interval two-alternative forced choice paradigm and the PSI method (*Kontsevich and Tyler, 1999*; *Prins, 2013*) that were implemented in the Palamedes toolbox 1.5.1 (*Prins, 2016*). In each trial, participants were presented with two pieces of white noise of 1 s with an inter-piece interval randomly sampled between 0.5 s and 1 s. The target tone was randomly inserted into one of the two noise pieces from 0.2 to 0.8 s after the noise onset. The white noise and the target tone were generated following the same procedures described above. The participants chose which noise piece included the target tone by pressing one of two buttons after the second noise piece was over. Visual feedback was provided to indicate whether the participants made the correct response in each trial, so that the participants could learn this tone-detection task and show stable performance in the following sessions. To get robust threshold measurements, we measured each participant's threshold twice. In the first measurement, the starting point of the SNR was set to –5 dB and the threshold was measured in 40 trials. In the second measurement, the starting point of the SNR was the threshold measured from the first measurement. Every 15 trials, there was a catch trial with a SNR of 5 dB to remind the participants of the target tone. The threshold measured from the second time was selected as each participant's threshold. The threshold here represented 75 percentage correct threshold.

The auditory stimuli were delivered through plastic air tubes connected to foam ear pieces (E-A-R Tone Gold 3 A Insert earphones, Aearo Technologies Auditory Systems).

## EEG recording

EEG data were recorded using an actiCAP 64-channel, active electrode set (10–20 system, Brain Vision Recorder, Brain Products, brainproducts.com) with a 0.1 Hz online filter (12 dB/octave roll-off). There were 62 scalp electrodes, and their impedances were kept below 5 kOhm. The first 18 participants of 27 valid participants were recorded at a sampling rate of 500 Hz with one electrode (originally, Oz) placed on the tip of the nose as the reference channel. The other 9 participants were recorded at a sampling rate of 10,000 Hz with one electrode placed at Oz as the reference channel. The impedance of the reference channel was kept below ~10 kΩ. This difference in EEG recording settings was caused by miscommunication between the experimenter and the research assistants who helped conduct EEG recording. To make the EEG data compatible between the two recording settings, in the following data analyses, EEG recordings were offline referenced to the average of activity at all scalp electrodes. The recording sampling rate difference should not matter here, as we later applied

an off-line low-pass filter of a cut-off frequency of 30 Hz and downsampled all the data to 100 Hz. Furthermore, the following major analyses included baseline corrections based on pre-stimulus activities in individual participants' data and focused on modulations of EEG power (a second-order statistics), the EEG reference differences between participants should not affect our results.

## Behavioral data analysis

Tone detection performance was analyzed in MATLAB 2016b using the Palamedes toolbox 1.5.1 under the framework of signal detection theory. In each block, we treated the trials with a tone embedded as the 'target' trials and the trials without a tone as the 'noise' trials. For each temporal location in each block, we derived a hit rate from the 'target' trials ('yes' when there was a tone) and a false alarm rate from the 'noise' trials ('yes' where there wasn't a tone). We then calculated a d-prime value using the hit rate and the false alarm rate assuming an independent observer model and an unbiased observer. To avoid infinite d-prime values, a half artificial incorrect trial were added when all 'target' trials were correct; a half artificial correct trial was added when all 'noise' trials were incorrect (*Hautus et al., 2021*). Note that the 'target' trials with the target tone appeared at one of the three temporal locations shared the same 'noise' trials, so the d-prime values and the criteria derived at each of the three temporal locations were not independent and the differences between the temporal locations were dominated by the hit rates.

Raw reaction time data were used in the data analysis. The reason that we did not take a log transformation of reaction time is that previous literature demonstrated that a log transformation of reaction time is not necessarily beneficial (*Harald Baayen and Milin, 2010*). Nevertheless, we conducted statistical tests on both log-transformed and raw reaction time data and found that the conclusions of the statistical tests remained the same.

## EEG preprocessing

EEG data analysis was conducted in MATLAB 2016b using the Fieldtrip toolbox 20181024 (*Oostenveld et al., 2011*) and the wavelet toolbox. EEG recordings were off-line referenced to the average of activity at all scalp electrodes. Raw EEG data were first filtered through a band-pass filter from 1–30 Hz embedded in the Fieldtrip toolbox (a 'fir' two-pass filter with an order of 4). The cutoff frequencies of 1 Hz and 30 Hz were selected here because previous literature showed that effects of temporal anticipation in the auditory modality were primarily observed in the delta band (1–4 Hz) and the beta band (12–30 Hz) (*Arnal et al., 2015*; *Morillon and Baillet, 2017*).

Continuous EEG data were first divided into 6 s epochs, including a 2 s pre-stimulus period, a 2 s stimulus period, and a 2 s post-stimulus period for each trial. By preserving long periods before and after the stimulus onset and offset, we had enough data to calculate low-frequency EEG power (e.g. at 1 Hz) to conduct the following baseline corrections and to remove filter artefacts in the beginning and the end of the EEG signals of each trial. The epoched trials were then visually inspected, and those with artifacts such as channel jumps and large fluctuations were discarded. An independent component analysis was applied over all the trials pooling over all the blocks in each session and was used to correct for artifacts caused by eye blinks and eye movements. After preprocessing, up to 10 trials were removed for each of the four conditions. Baseline in each trial was corrected by subtracting out the mean of –1 s to 0 s.

We conducted various following analyses on the Cz channel as past research shows that auditory-related and anticipation-related effects can be well captured by the Cz channel. After we found certain effects at the Cz, for visualization, we conducted the same analyses at other channels and plotted topographies.

## Event-related potential (ERP) to tones

We calculated evoked responses to the target tone at each temporal location. The baseline of each trial was corrected as described in *EEG preprocessing*. We chose not to conduct baseline correction again for ERPs to the target tone at different temporal locations, because the target tone at 500 ms was affected by the noise onset response whereas the tones at 1000 ms and 1500 ms were not. Re-correcting baselines for ERP to the target tone at each temporal location may introduce artefacts. We focused on the period of ERPs from 0 ms to 500 ms after the tone onset in the following statistical analyses.

We pooled trials across all three blocks and calculated ERPs. This pooling procedure guaranteed that we had enough trials to detect differences of ERPs between three temporal locations with sufficient statistic power. We conducted a permutation-based cluster repeated measures ANOVA (rmANOVA) analysis over the three temporal locations (*Maris and Oostenveld, 2007*), in which we shuffled the labels on the temporal locations in each participant's data. By doing this, we constructed a null hypothesis, which was that the ERPs to the target tone at three temporal locations do not differ from each other. During each time of shuffling, we constructed a new dataset and conducted a one-way rmANOVA test at each time point with Temporal location as the main factor. We identified clusters of connected significant time points (>2 time points) at a two-sided alpha level of 0.05 and calculated a cluster-level $F$ value by adding up F values within each cluster across time points. The biggest cluster $F$ value was selected as the cluster-level $F$ value for each time of shuffling. We conducted this procedure 1000 times and derived a distribution of cluster-level $F$ values, from which a threshold of cluster-level $F$ values at a one-sided alpha level of 0.05 was derived (the 950[th] cluster $F$ value). We then derived significant clusters from our empirical data and determined which cluster was significant by comparing their cluster-level F values with the threshold of cluster-level $F$ values. Then, we conducted paired t-tests, as post-hoc tests, within the temporal regions of the significant clusters to compare ERPs between each pair of temporal locations. Adjusted FDR was used for multiple comparison correction.

## Time-frequency decomposition of EEG signals

To extract time-frequency information, single-trial data in each EEG channel were transformed using the Morlet wavelets embedded in the Fieldtrip toolbox, with a frequency ranging from 1 to 40 Hz in steps of 1 Hz. Note that we did not intend to analyze the signals above 30 Hz since we applied a low-pass filter at 30 Hz. We simply applied our previous routine analysis parameters here.

To balance spectral and temporal resolution of the time-frequency transformation, from 1 to 40 Hz, the window length increased linearly from 2 circles to 20 circles. Phase and power responses (squared absolute value) were extracted from the wavelet transform output at each time-frequency point. The temporal resolution was set as 100 Hz and the power responses from –1 s to 3 s were selected for the following analyses.

In each condition, we first averaged power values over all the trials and then calculated a baseline by averaging the mean power values from –0.75 s to –0.25 s. The averaged power after the noise onset (0 s) was normalized by dividing the baseline and taking logarithms with base 10, and then was converted into values with unit of decibel by multiplying 10.

As the power was first calculated on the single-trial level and was then averaged across trials, the resulting power was sometimes called 'induced power;' in contrast, 'evoked power' is calculated by first averaging trials in the temporal domain and then conducting the time-frequency decomposition (*Makeig et al., 2004*). It has been shown that the evoked power reflects phase-locked temporal coherence between trials because the averaging procedure was taken before time-frequency decomposition (*Teng and Poeppel, 2020*). However, this labeling of 'evoked' and 'induced' is not entirely correct because the averaging in time does not necessarily average out non-phase locked activity (*Mazaheri, 2022*). Thus, we directly analyzed inter-trial phase coherence below to quantify the phase-locked neural components across trials (*van Diepen and Mazaheri, 2018*). As a result, we simply use 'power' to indicate our calculation of power here and analyzed the phase domain directly to represent time-locked neural activity that is arguably related to 'evoked power'.

## Power modulation spectrum (PMS)

We first quantified how the power was modulated by temporal anticipation for the temporal locations over the entire trials by calculating PMS. We also conducted the corresponding analysis on the phase (see *ITPC*). If the participants could register the three temporal locations of the target tone in their memory and actively anticipate the tagged temporal locations every 500 ms, we would find a neural signature around 2 Hz reflecting such sequential anticipating processes even when only the white noise was presented in each trial.

Note that this method is different from the method of directly applying 2D-FFT on acoustic spectrograms to derive modulation power spectrum (MPS) (*Elliott and Theunissen, 2009*). To avoid this confusion, we intentionally named our calculation as PMS.

We focused on the condition in which only the white noise was presented and pooled all the no-tone trials over three blocks. We selected the neural data between 0.25 s and 1.75 s after the noise onset to avoid effects of onset and offset responses. We did not define any EEG frequency bands but used a data-driven method to find significant modulation effects on the power from 1 Hz to 30 Hz.

We first applied a high pass filter at 0.5 Hz on the power at each frequency over entire trials (–1 s to 3 s) so that we removed neural components over a duration larger than the length of the white noise pieces (2 s). We next applied a Hamming window of 1.5 s (151 points) on the selected time range, 0.25 s to 1.75 s after noise onset, at each frequency of power and conducted the Fourier transform with a length of 500 points (151 data points padded with 349 zero points). The spectral resolution of the resulted spectrum of the power was 0.2 Hz. The PMS derived have two spectral dimensions: one dimension was the frequency of power ('power frequency') and the other dimension was the 'modulation frequency' of modulations of power. As the power was normalized to have a unit of dB, the unit of PMS was also dB since the Fourier transform only includes linear operations.

Here, we laid out two further constraints due to the parameter selections in the above analyses used to derive the PMS. First, the duration of power selected for the PMS analyses was 1.5 s long (0.25 s to 1.75 s after noise onset), which meant that the lowest possible modulation frequency representing more than two cycles of power fluctuations was 1.33 Hz. At modulation frequencies less than 1.33 Hz, one could not determine whether the modulation components represented by the frequencies are rhythmic or belong to a duty cycle of an even lower frequency, or simply indicate a trending signal. Therefore, there is Trend bound at 1.33 Hz along the axis of modulation frequency. Neural components above 1.33 Hz can be considered to have rhythmic modulation components; neural components below 1.33 Hz likely represent trending responses from 0.25 s to 1.75 s. Second, the power at one power frequency can only fluctuate at a modulation frequency less than one-half of the power frequency according to the Nyquist-Shannon sampling theorem. The power at one power frequency can be viewed as a signal carrier; the power modulation envelope of this carrier can only have a modulation frequency as high as half of the carrier frequency. For example, the modulation frequency derived from the power at the power frequency of 1 Hz can only be as high as 0.5 Hz in theory. If we examine the modulation frequency of 2 Hz on EEG power, we have to look at the EEG power at least above 4 Hz of the power frequency. Hence, there is a sampling bound. We indicated those two constraints, Trend bound and Sampling bound, in *Figure 2B*.

Note that the PMS was not affected by different baseline correction procedures used in *Time-frequency decomposition of EEG signals*. The reason is that we took a log transformation of EEG power after the division between the power of the stimulus period and the baseline power. This is equal to taking a log of the power of the signals and a log of the baseline power separately and then subtracting the two values linearly. The baseline power did not change in time and, after the Fourier transform was applied, was separated out from the EEG power of the signals as a DC component in the modulation spectra (the component at 0 Hz).

## Surrogate test on PMS

To determine the spectral regions of PMS showing significant modulation strength, we conducted a surrogate test in the spectral domain at the group level. The rationale is that, as all the participants were doing the same task, neural dynamics reflecting the underlying cognitive processes should be similar across individual participants. At the group level, we should be able to identify neural dynamics that are consistent across the participants. Hence, the null hypothesis here is that no consistent temporal modulation of power at any power frequency exists across the participants.

To construct the null hypothesis, after we applied the Fourier transform on the EEG power at each power frequency and obtained the Fourier coefficients (complex numbers), we multiplied a complex number of a random phase and of a norm of 1 with the Fourier coefficient at each modulation frequency. By doing this, we disrupted the phase of each modulation frequency at each power frequency but preserved the modulation strength in each participant's data – this is equal to disrupting temporal dynamics. We then averaged the surrogated data over all the participants in the spectral domain and derived a new group-level PMS. We conducted this procedure 1000 times and constructed a null distribution of PMS at each power frequency and each modulation frequency, from which we derived a threshold of a one-sided alpha level of 0.05.

The reason that we chose the surrogate test in the spectral domain but not in the temporal domain is that we would like to specifically examine the temporal alignment of modulations on the power across the participants while keeping the modulation strength unchanged. To conduct a surrogate test in the temporal domain, we would have to change the time range of individual participants' EEG data by shifting the time range of interest forward or backward in each surrogate. The shifted time range may include noise onset/offset responses or data from the fixation period, so both the modulation strength and phase of power in the shifted time range are changed and are different across participants.

We selected four significant clusters (>2 spectral bins) that were robustly shown in each time of the surrogate test with high modulation power (Clusters 1, 2, 3, and 4 in *Figure 2B*). Other significant modulation bins were also observed, but they either had low modulation strength in the PMS or fell below Sampling bound (likely caused by the ripple artefacts of the Fourier transform). The power frequency ranges/the modulation frequency ranges for the four clusters are as follows: Cluster 1, 1–4 Hz/0–1.33 Hz; Cluster 2, 22–27 Hz/0–1.33 Hz; Cluster 3, 9–14 Hz/1.8–2.8 Hz; Cluster 4, 24–26 Hz/1.8–2.8 Hz.

The modulation frequency range for each cluster was not selected strictly according to the significant spectral bins in the PMS of *Figure 2B*. The significant modulation frequencies across different power frequencies in one cluster were not the same, so it was implausible. to filter the power strictly according to the significant frequency bins when the temporal domain of the power was investigated. Defining a 'square' region of interest in the PMS for each cluster facilitated filtering procedures in the following analyses of extracting temporal dynamics; simplistic filtering procedures avoided potential artefacts introduced by filtering each power frequency using a different modulation frequency range. Therefore, we chose the spectral bins for the clusters in the following manner: (1) the significant modulation frequencies of Cluster 1 and 2 were below Trend bound and potentially included trending responses that lasted longer than the EEG data period (1.5 s) included in the PMS analyses, so we used 0 Hz as the low cutoff modulation frequency and set the high cutoff frequency as 1.33 Hz; (2) Cluster 3 and 4 were centered around 2 Hz of the modulation frequency and we selected the same modulation frequency range for both the clusters, so that later we could compare temporal dynamics of power within the same modulation frequency range.

The PMS analyses were first conducted at Cz to investigate whether our hypothesized neural phenomena (*Figure 1D*) could be observed. We later examined other EEG channels and created a topography for each cluster to reveal distributions of modulation strength across EEG channels. We filtered the power according to each defined cluster and conducted the surrogate test described above on each EEG channel. We derived thresholds at one-sided alpha level of 0.05 to determine whether the power modulation was significant for each cluster on each EEG channel, which was indicated by the green dots in *Figure 2C*. When we plotted the topographies in *Figure 2C*, we normalized the modulation strength by subtracting the mean of the null distributions out of the empirical modulation strength so that the distribution of the modulation strength across EEG channels was highlighted in the topographies.

## Temporal dynamics of clusters and power coupling

The above analyses in the spectral domain did not reveal temporal details of the power fluctuations, so we next investigated temporal dynamics of power within those four clusters. We first averaged the power within the power frequency range of one ROI/cluster and then filtered the averaged power using a band-pass filter (for Cluster 3 and 4, 1.8–2.8 Hz) or a low-pass filter (for Cluster 1 and 2, <1.33 Hz) (two-pass Butterworth filter of an order of 2 implemented in Fieldtrip).

We next checked whether each cluster was independent from or coupled with other clusters. By doing this, we can validate previous findings on the coupling of neural signals between the delta and beta bands in prospective timing paradigms and validate our data-driven procedures here. Considering that the Fourier transform projects signals onto an orthogonal space of sinusoid waves, a complex neural signal may be broken into several clusters. For example, Clusters 3 and 4 are probably in a harmonic relationship: Cluster 4 is likely the harmonic of Cluster 3 as the power frequency of Cluster 4 (24–26 Hz) is approximately twice of the power frequency of Cluster 3 (9–14 Hz). If this is the case, Clusters 3 and 4 should be coupled.

We calculated phase locking values (PLV) on the power across participants to measure the coupling between clusters at each time point. We applied the Hilbert transformation on the power of each cluster and then derived the phase value at each time point by calculating angle of the Hilbert coefficient. The phase difference between each pair of clusters for each participant was then used to calculate PLV across all the participants using the following formula, in which $S$ denotes the total number of participants, $s$ the participant number, $c$ the cluster, $a$ and $b$ numbers of two different clusters, $t$ the time point, and $\phi$ the phase at each time point:

$$PLV_{c_{ab},t} = \left| \sum_{s=1}^{S} e^{(\phi_{s,t,c_a} - \phi_{s,t,c_b}) * i} \right| / S \tag{1}$$

If two clusters are consistently coupled across participants, the phase differences across all the participants at one time point should concentrate around a specific phase value – a high PLV value (low circular variance across participants). Accordingly, we constructed a null hypothesis to determine the significant time points of PLV values through a surrogate test in the spectral domain. In each participant's data, we reset the phase of each cluster at one time point randomly by multiplying a complex number of a random phase and a norm of 1 with the Hilbert coefficient. By doing this, we disrupted the phase series of each cluster independently and hence the coupling strength between two clusters. We then calculated new PLV values of each pair of clusters across all the participants. We conducted this procedure 1000 times and constructed a null distribution of PLV at each time point, from which we derived a threshold of a one-sided alpha level of 0.05 for each cluster pair.

## Merge Clusters 2, 3, and 4 into Cluster 234 and further analyses

From *Figure 2D*, we observed that Cluster 1 slowly changed along the entire trials and showed a global trajectory capturing the length of the noise piece, whereas the dynamics of Clusters 2, 3, and 4 seemingly did not correspond to the temporal structure in each trial. However, a peak after stimulus onset was observed in Cluster 2 around the first temporal location, Cluster 3 and 4 were coupled and both showed peaks around the second and the third temporal locations (*Figure 2D*, *Figure 2—figure supplement 1C*), and Cluster 2 and 4 are within the same EEG power frequency band (*Figure 2B*). Overall, the above observation gave us a clue that Clusters 2, 3, and 4 may collectively capture the three temporal locations of the target tone. We tried averaging the EEG power over Clusters 2, 3, and 4 and merged them into Cluster 234. Surprisingly, the merged EEG power dynamics showed a salient neural signature marking the temporal locations, even if only the white noise was presented. By plotting Cluster 1 and Cluster 234 together, it can be clearly seen that Cluster 1 in the delta band reflected the global trial structure (the length of noise pieces) and that Cluster 234 captured the local temporal locations of the target tone. As we aimed to find neural signatures tracking the temporal locations and similar findings in the delta band, corresponding to Cluster 1, have been reported before to encode event duration or fixed intervals (*Breska and Deouell, 2017*; *van Wassenhove and Lecoutre, 2015*), we next only focused on Cluster 234.

The dynamics of Cluster 234 contains a downward linear trend and a rhythmic component (*Figure 2*). To single out the rhythmic component, we fitted a line to Cluster 234 from –1 s to 0 s and subtracted out the fitted line. The resulted neural power modulation, Corrected cluster 234, clearly showed peaks after each temporal location.

Please note that we altered the range used to fit the baseline for Cluster 234, from –1 s to –0.3 s and from –1 s to –0.2 s, in order to avoid the confounding factor that the power of the baseline close to 0 s may contain stimulus-evoked signals post the onset of noise at 0 s. However, the different baseline ranges did not yield significant differences, likely due to the relatively stable trend of the baseline between –1 s and 0 s, as depicted in *Figure 2E*. Furthermore, there was no 'onset response' after the noise onset for Cluster 234 as illustrated in *Figure 2E*. The power dynamics remained 'stable' until after 0.5 s, when the first peak of Cluster 234 emerged. In instances where onset responses occur immediately post-stimulus onset, including neural signals close to 0 s in the baseline could be problematic, but this was not the case for Cluster 234. Therefore, we assert that using the range from –1 s to 0 s to derive a baseline for correcting Cluster 234 did not impact subsequent analyses.

We then individually fitted a Gaussian function to Corrected cluster 234 from each tone onset position to 0.5 s afterwards. From the fitted Gaussian curves, we derived the peak latency (the position of

the mean in the fitted Gaussian curve) and the standard deviation (SD). We averaged each neural peak of Corrected cluster 234 within one SD centered around the peak latency (–0.5 SD + tone location to 0.5 SD + tone location) for each participant and conducted a one-way rmANOVA with Temporal location as the main factor.

To visualize the neural tracking of the temporal locations across all EEG channels, we calculated the modulation strength of Corrected cluster 234 from 0.4 s to 1.9 s at each channel (SD over this period of signal) and plotted the topography of the SD. The time range was chosen based on the first zero-crossing point after the noise onset and the time point after the third peak showing the lowest power magnitude. The first zero-crossing point indicated the rising of the EEG power above zero and the second time point the EEG power falling towards zero. The SD over this time range represented the degree of EEG power fluctuations, which can be seen as the amplitude summed over all the frequency components of Corrected cluster 234 within this time range.

## Block number on power modulation of corrected cluster234

We next examined how learning from the first to the third block affected the sequential temporal anticipation. The participants probably could not precisely anticipate the temporal locations of the target tone in the first block, but gradually learned the exact temporal locations in the following blocks. We expected to find that the magnitude of Corrected cluster 234 increases from the first block to the third block.

We extracted the power from the spectral regions defined in Clusters 2, 3, and 4 in the data from each block and merged the three clusters into Cluster 234 in each block. As described in the above analyses, the PMS calculation was not affected by differences of baseline corrections, so the baseline differences between the blocks did not contribute to the following analyses. To correct Cluster234 in each block, we adopted the fitted baseline from the above analyses on the trials pooled across all three blocks, so that Corrected cluster 234 in each block went through the same baseline correction procedure. Similarly, following the above procedures, we fitted Gaussian curves to peaks after each temporal location in each block (*Figure 3*, *Figure 3—figure supplement 1*) to extract peak magnitudes. We conducted a two-way rmANOVA on the peak magnitudes with Temporal location and Block number as the main factors to examine the learning effect (*Figure 3B*).

## Correlation between behavioral performance and Corrected cluster234

We correlated d-prime values with the peak magnitude of Corrected cluster 234 in each block to examine whether the identified neural signature coding temporal locations in the no-tone trials correlated with the tone detection performance in the tone trials (*Figure 3*, *Figure 3—figure supplement 1*). It is possible that, since the power of Corrected cluster 234 was derived in the no-tone trials, the power peaks may be caused by false alarms – the participants thought they heard a tone even though there was no tone, so we then correlated the power peaks with the false alarm rates.

## Quantify peak latency change of Corrected cluster 234 across blocks and temporal locations

The peak latencies of Corrected cluster 234 decreased from the first temporal location to the third and were modulated by the block number (*Figures 2G and 3A*). We here verified this observation quantitatively. One issue here was that the peak latencies determined by the Gaussian fits were derived from the group-averaged power; individual participant's data were noisy so that the Gaussian fits to each participant's neural power were not accurate. This prevented us from drawing conclusions by conducting parametric statistic tests over participants (e.g. rmANOVA). To circumvent this issue, we resorted to group-level permutation tests.

We first used a permutation test to examine one observation from *Figure 3A*: the peak latencies were different between blocks. We constructed a null hypothesis that, at each temporal location, there is no peak latency difference between blocks. One reason for constructing this null hypothesis first was that the null distribution constructed at each temporal location could be viewed as a distribution that averages out the factor of the block number. Later, by comparing three null distributions of peak latency in the three temporal locations, we could determine whether the peak latencies of Corrected cluster 234 varied with the temporal locations.

To construct the null hypothesis of the peak latency at each temporal location across blocks, we permuted the block label in a way that each participant's data in each block were randomly assigned with a new block number. We grouped the permuted data into three new blocks at each temporal location and derived Corrected cluster 234 in each new block. We fitted a Gaussian curve to each power peak to derive the peak latency by following the above procedure. We conducted this permutation procedure 1000 times and derived nine null distributions of peak latencies (three temporal locations * three blocks). As the block labels were shuffled and were not meaningful anymore in the null distributions, we pooled the null distributions over three blocks and had one null distribution of 3000 samples for each temporal location. We derived thresholds at a two-sided alpha level of 0.05 (the 76[th] latency and the 2925[th] latency) from each null distribution (*Figure 3D*). The thresholds derived can be viewed as significant boundaries – if a peak latency from one block of the empirical data falls within the boundaries, the peak latency does not significantly differ from peak latencies of the other blocks; if a peak latency is outside of those significant boundaries, it is then significantly different from the other peak latencies. The boundaries, in a sense, represented an interval that the peak latencies of the three blocks are uniform, statistically speaking, so we called them as the 'uniform' bound and the permutation test as the 'uniform' test.

The null distributions of peak latency at each temporal location represented distributions of possible peak latencies without considering the factor of the block number, as each distribution was derived from random combinations of the participants' data drawn from the three blocks. Therefore, by randomly selecting a sample from each null distribution and examining how the peak latencies changed from the null distribution of the first temporal location to the third, we could further derive a distribution of the peak latency change that could be used to determine whether the peak latencies significantly varied with the temporal locations. One simple way to quantify the latency change was to fit a straight line to the peak latencies with the temporal location as the independent variable of values from 1 to 3 and the latency as the dependent variable. If the slope is significantly negative, it can be concluded that the peak latency decreases significantly with the temporal locations. Following this logic, we fitted a line to peak latencies randomly selected from the three null distributions and constructed a distribution of slope of 3000 samples. We derived a threshold at a two-sided alpha level of 0.05 from the slope distribution and compared the thresholds with zero to determine whether the peak latencies significantly shortened from the first temporal location to the third (*Figure 3F*).

## Modulation spectrum analysis on inter-trial phase coherence (ITPC)

Besides the power measurement of neural signals, the anticipation for a future event may cause neural phases to concentrate around a certain phase value, as suggested in previous literature (*Breska and Deouell, 2017*; *Samaha and Postle, 2015*). If this is the case, we would expect that the neural phases in the no-tone trials were consistently locked to the tagged temporal locations and hence the variance of neural phases across trials decreased before or after the temporal locations. Accordingly, a 2 Hz component should be observed in the ITPC that measures consistency of neural phases across trials. Here, we conducted the modulation spectrum analysis on ITPC over the no-tone trials pooled from the three blocks.

Note that ITPC also reflects routinely defined 'evoked power', as it has been shown that the evoked power reflects phase-locked temporal coherence between trials because the averaging procedure was taken before time-frequency decomposition (*Teng and Poeppel, 2020*). However, this labeling of 'evoked power' is not entirely correct because the averaging in time does not necessarily average out non-phase locked activity (*Mazaheri, 2022*). Here, we directly analyzed inter-trial phase coherence to quantify the phase-locked neural components across trials. In this study, we simply use 'power' to indicate our calculation of power here and analyzed the phase domain directly to represent time-locked neural activity that is arguably related to 'evoked power'

The Fourier coefficients were derived in *Time-frequency decomposition of EEG signals*. We derived the neural phase at each time-frequency bin by calculating the angle of the Fourier coefficient. The formula of ITPC calculation at a time–frequency bin is shown below with $N$ denoting the total number of the trials, $n$ the trial number, $f$ the frequency point, $t$ the time point, and $\phi$ the phase value at each time-frequency bin:

$$\text{ITPC}_{t,f} = \left| \sum_{n=1}^{N} e^{\phi_{n,t,f}*i} \right| / \text{N} \tag{2}$$

We used the valid trials across the three blocks after *the EEG preprocessing* to calculate ITPC. We selected the time range between 0.25 s and 1.75 after the noise onset to avoid effects of onset and offset responses of the white noise. Then, following the procedures in *PMS*, we conducted modulation spectrum analyses as well as surrogate tests on ITPC values to determine significant modulation clusters. Only one salient cluster was shown below Trending bound and, in the theta and delta bands along the axis of ITPC frequency (*Figure 2—figure supplement 2*). Since no 2 Hz modulation component was found in the ITPC values, showing that the phase-locked neural responses did not code the temporal locations, we did not conduct any further analyses on the neural phases.

## Train recurrent neural network models to accomplish the tone-detection task

We chose to train CRNNs with mini-batch gradient descent learned by backpropagation. The structure of CRNNs is defined as the following equation:

$$\tau \frac{dx_i}{dt} = -x_i + \sum_{j=1}^{N} W_{ij}^{rec} r_j + \sum_{k=1}^{N_{in}} W_{ik}^{in} u_k + \xi_i \tag{3}$$

$$r_i = \left[ x_i \right]_+ \tag{4}$$

$$\mathfrak{z}_l = \sum_{i=1}^{N} W_{li}^{out} r_{i,T} \tag{5}$$

The number of hidden units, $N$, was 100. We tested CRNNs with fewer hidden units (64 and 32) and found out that those CRNNs could not be trained to achieve human-level performance when the external noise level was high in our settings. The time step size, $\Delta t$, was 25 ms (a sampling rate of 40 Hz), which was sufficient to capture the dynamics at 2 Hz shown in our neural findings and made the training efficiently. $\tau$ is the time constant of the network units and was set to 100 ms. $u$ is the input vector, so $k$ can only be 1. $\mathbf{W^{in}}$ is a *1×N* matrix of connection weights from the inputs to $N$ network units. $\mathbf{W^{rec}}$ is an N×$N$ matrix of recurrent connection weights between network units. $\xi$ are $N$ independent Gaussian white noise processes with zero mean and unit variance that represent noise intrinsic to the network. The rectified linear activation function (ReLU) was applied on hidden units. $\mathbf{W^{out}}$ is an N×*2* matrix of connection weights from $N$ network units to the outputs (two choices: yes or no). The outputs, $\mathfrak{z}$, were generated after the input is over; the ending time point is indicated by $T$. The CRNN, a continuous-time formulation of a standard RNN, was converted to a discrete-time version through the Euler method for implementation.

The connections between hidden units were constraint according to Dale's rule: 80% of units were fixed to be excitatory units and 20% of units were inhibitory units (*Song et al., 2016*). We initialized the weights by generating random weights using the normal distribution with the mean of zero and the standard deviation (SD) of square root of 5, taking absolute values of all weights, multiplying 20% of weights with –1 to set them as inhibitory units, and equalizing the gain between excitatory units and inhibitory units. The diagonal of the weight matrix (self-connections of units) was fixed at 0. The internal noise level of $\xi$ was set to 0.01, so that we could focus on studying the effect of noise level of inputs by varying the noise level of inputs systematically.

The inputs fed to CRNNs were constructed in NeuroGym (https://neurogym.github.io/) to mimic the trial structure of the experiment (*Figures 1A and 4A*). The magnitude of the fixation period in each trial was set as zero with additive Gaussian noise of the mean of zero and SD of 0.05. The length of the fixation period varied from 1000 ms to 1500 ms in different batches during training in a step size of 25 ms. For example, in one batch, the fixation period was 1000 ms; in another batch, the fixation period became 1175 ms. The stimulus period was 2 s and the noise was Gaussian noise with the mean of 1 and various levels of SD according to the following training designs. The target magnitude was set as 1 and was added to the stimulus period at either 500 ms, 1000 ms, or 1500 ms. The CRNNs made the decision by the end of the stimulus period to answer whether there was a target in a trial – a yes-no response. We did not simulate the reaction time but only aimed to simulate the procedure of

human participants detecting a tone and indicating their decision by pressing a button as soon as the white noise was over. The loss was calculated only by the end of trials using cross-entropy.

During training CRNNs, each mini-batch included 16 trials with each trial equally likely coming from each of four conditions (three tone trial conditions and one no-tone trial condition) (*Figure 4A*). We set the maximum training sessions to 40,000 and terminated the training if the mean accuracy over the last 100 mini batches was above 99 percentage correct. We used ADAM (*Kingma and Ba, 2014*) during training and scheduled the learning rate to drop in three steps: 0.001 for 10000 sessions, then 0.0001 for 20000 following sessions, and 0.00001 for the last 10000 sessions. All the training procedures were implemented in PyTorch 1.7.0 (*Paszke et al., 2019*).

The main parameter we varied during training and testing CRNNs was the external noise level of the stimulus period. The external noise was added to the stimuli and was shared across all the hidden units of CRNNs. We trained five CRNNs at five external noise levels, separately: 0.01, 0.05, 0.1, 0.15, and 0.2 SD, and tested each trained CRNN at 7 noise levels: 0.01, 0.05, 0.1, 0.15, 0.2, 0.25, and 0.3 SD.

## CRNN behavioral performance, hidden state dynamics, and modulation spectra

We trained five CRNNs at different noise levels, respectively, and tested them across all the seven testing noise levels. We generated 27 sets of behavioral data at each testing noise level to match the number of recruited human participants. Following the same procedure in *Behavioral data analysis*, we calculated d-prime values from CRNN outputs. Then, we selected one testing noise level for each of the five CRNNs according to how close the group-level d-prime value derived at this testing noise level matched the group-level mean of the human behavioral data averaged over the three temporal locations in *Figure 1—figure supplement 1C*.

We fed each of the five trained CRNNs with 40 no-tone trials at the chosen testing noise level and extracted activities of their hidden units. As neural signals recorded by EEG reflect summarized neural activities over both excitatory and inhibitory synapses, similarly, we averaged the CRNN hidden state activities over both excitatory and inhibitory units. We analyzed the averaged hidden state activities following the procedure used in calculating the modulation spectrum of EEG signals. We selected the averaged hidden activity of each trial from 0.25 s to 1.75 s after noise onset and applied the Fourier transform, and then extracted the amplitude of the Fourier coefficients for each trial and averaged the modulation spectra over all the 40 no-tone trials to generate an averaged modulation spectrum of the hidden activities. We found that the CRNN trained at the noise level of 0.15 SD and tested at the noise level of 0.2 SD showed the largest 2 Hz modulation strength (*Figure 4C*), so we selected this CRNN for the subsequent investigations as it both matched the human behavioral performance and the neural findings (*Figure 2*). The hidden activities of CRNNs shown in *Figure 4D and E* were derived from averaging hidden activities over 40 no-tone trials.

## Train and test CRNNs with a uniform probabilistic distribution of tone position from 0.25 s to 1.75 s

We trained a CRNN with a new trial structure in which the target tone can appear at any time from 0.25 s to 1.75 s with an equal probability (*Figure 4F*), so that the newly trained CRNN can be compared with the CRNNs trained with a target tone appearing at one of the three temporal locations. The motivation here is twofold: (1) If the 2 Hz modulation component was driven by specific temporal locations, the CRNN trained with the new trial structure should not contain the 2 Hz modulation component; (2) We aimed to test tone detection performance of the CRNN trained with the three temporal locations at other time points.

In the original trial structure for training CRNNs (*Figure 4A*), 75 percent of trials had the target tone embedded and the other 25 percent of trials were no-tone trials. Accordingly, here, we fixed 25 percent of trials to be no-tone trials, and, in the other 75 percent of trials, a target randomly appeared any time from 0.25 s to 1.75 s. In each training batch, the trial type was randomly sampled with a probability of 0.75 to be tone trials and 0.25 to be no-tone trials. We chose the training noise level to be 0.15 SD. The other parameters and procedures for training CRNNs were the same as in the previous CRNN training procedures. After 40,000 batches, the training accuracy reached 0.97.

We tested both the newly trained CRNN and the CRNN trained at 0.15 SD with three temporal locations at a testing noise level of 0.2 SD. We chose a different way to analyze the tone detection performance, in contrast with the above behavioral analyses (*Figure 4B*). We independently generated 40 no-tone trials and 40 tone trials with the target at one time point, and asked CRNNs to generate outputs. For each time point, we measured a hit rate from the tone trials and a false alarm rate from the no-tone trials and calculated a d-prime value. This d-prime calculation was conducted from 0.25 s to 1.75 s with a step of 0.025 s (*Figure 4G*).

## Perturb E/I ratio of the CRNN trained at the noise level of 0.15 SD

To account for the neural power precession observed in *Figure 3D*, we perturbed excitation-inhibition balance of trained CRNNs. The perturbed CRNNs may generate new hidden dynamics that potentially capture the phenomenon of the shortened peak latencies of Corrected cluster 234.

We conducted the perturbation on the CRNN trained at the noise level of 0.15 SD with three temporal locations and derived the hidden activities at the testing noise level of 0.2 SD, as the above analyses showed that this CRNN captured the neural dynamics successfully (*Figure 4C–E*). To perturb the excitation-inhibition balance, we changed the weights of excitatory and inhibitory units. For example, we increased the weights of excitatory units by 10 percent so that the excitatory units sent stronger projections to all units. However, only increasing excitatory unit weights would disturb the CRNN and potentially drive it to a chaotic state or to a silent state. Hence, we changed the weights of excitatory and inhibitory units by the same amount but in opposite directions. For example, when the perturbation was 10 percent, we increased the excitatory weights by 10 percent and, correspondingly, decreased the inhibitory weights by 10 percent.

We perturbed the excitation-inhibition balance from –40 percent (negative gain for excitatory weights and positive gain for inhibitory weights) to 40 percent with a step of 5 percent. After we perturbed CRNNs, we asked the CRNNs to generate behavioral outputs and calculated d-prime values. Meaningful positive d-prime values were generated when the perturbations were –10, –5, 5, 10, and 15 percent, so we only analyzed the CRNNs under those five perturbations. We derived the averaged hidden activities of the perturbed CRNNs from 40 no-tone trials at a noise level of 0.2 SD (*Figure 4H*) and calculated d-prime values at three temporal locations from 27 simulated datasets following the procedure used in analyzing human behavioral data (*Figure 4I*).

## Acknowledgements

We thank Johannes Messerschmidt, Anna-Lena Possmayer, and Dominik Thiele for their assistance with data collection. We thank Xing Tian and Nai Ding for their comments on a previous version of the manuscript. We thank David Poeppel for support. This work was supported by the Max-Planck-Society. RYZ was supported by National Natural Science Foundation of China (32441102) and Shanghai Municipal Education Commission (2024AIZD014). XT was supported by Improvement on Competitiveness in Hiring New Faculties Funding Scheme, the Chinese University of Hong Kong (4937113).

## Additional information

### Funding

| Funder | Grant reference number | Author |
| --- | --- | --- |
| National Natural Science Foundation of China | 32441102 | Ru-Yuan Zhang |
| Shanghai Municipal Education Commission | 2024AIZD014 | Ru-Yuan Zhang |
| Chinese University of Hong Kong | 4937113 | Xiangbin Teng |
| Max Planck Society | | Xiangbin Teng |

The funders had no role in study design, data collection and interpretation, or the decision to submit the work for publication.

## Author contributions
Xiangbin Teng, Conceptualization, Data curation, Software, Formal analysis, Investigation, Visualization, Methodology, Writing – original draft, Project administration, Writing – review and editing; Ru-Yuan Zhang, Software, Formal analysis, Investigation, Visualization, Methodology, Project administration, Writing – review and editing

## Author ORCIDs
Xiangbin Teng  https://orcid.org/0000-0001-5360-4957
Ru-Yuan Zhang  https://orcid.org/0000-0002-0654-715X

## Ethics
The study was approved by the Ethics Council of the Max Planck Society (Approval No. 2017_12). All participants gave written informed consent prior to participation and consent to publish anonymized data. Participants were screened to ensure they were neurologically healthy and had normal hearing. All participants received monetary compensation for their time. The study was conducted in accordance with the Declaration of Helsinki.

Reviewer #1 (Public Review): https://doi.org/10.7554/eLife.99383.2.sa1
Reviewer #2 (Public Review): https://doi.org/10.7554/eLife.99383.2.sa2
Reviewer #3 (Public Review): https://doi.org/10.7554/eLife.99383.2.sa3

# Additional files

## Supplementary files
MDAR checklist

## Data availability
All data and code supporting this study are openly available on the Open Science Framework (OSF) at https://osf.io/8z9h6/. Further information and requests for resources should be directed to and will be fulfilled by the Lead Contact, Xiangbin Teng.

The following dataset was generated:

| Author(s) | Year | Dataset title | Dataset URL | Database and Identifier |
|---|---|---|---|---|
| Teng X | 2020 | Sequential Temporal Anticipation Characterized by Neural Power Modulation and in Recurrent Neural Networks | https://osf.io/8z9h6/ | Open Science Framework, 10.17605/OSF.IO/8Z9H6 |

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
