## [Editor Report · eLife assessment]

This **valuable** study provides insights into how the brain learns to better detect a target by predicting when the target may appear. Overall, **solid** evidence is provided that the power fluctuations of alpha- and beta-band oscillations can reflect the predicted occurrence time of the target, but some conclusions, especially ones related to the neural-network model and temporal gain control account, need further consideration. The study highlights an advanced EEG analysis approach as well as a close combination of human EEG analysis and computational modeling using recurrent neural networks.

---

## [Referee Report · Reviewer #1 (Public Review)]

Summary:

In this article, the authors investigated how the brain anticipates sequences of potential sensory events, using temporal predictability to enhance perception. To do so, they combined a tone detection task, electrophysiological recordings, and recurrent neural network models. The stimuli consisted of continuous white noise embedded with either a single tone presented at one of 3 equidistant (500ms) temporal locations, or no tone. The main analyses were carried out on no-tone trials, in which subjects only anticipated future events. First, a modulation power spectrum analysis revealed 4 frequency clusters, and a coupling analysis allowed the authors to group 3 of them together into cluster 234. The time course of the latter aligned with the temporal locations, reaching a local maximum following each of them. The power of cluster 234 during no-tone trials was positively correlated with behavioral performance (d') during tone trials, but not with false alarm rate. Then, the authors trained several continuous-time recurrent neural networks to model the experimental paradigm. After the networks were tuned to reflect the average d' of human subjects, a neural network analogue of EEG was extracted from the activity of neurons. The latter displayed a peak at 2Hz, its time course aligned with the temporal locations, reaching a local maximum both before and after each of them, and its d' score was higher for tones located at one of the temporal locations. A network trained with randomly occurring tones displayed no 2Hz activity and d' independent from tone location. Finally, the authors perturbed the excitatory/inhibitory ratio of neurons within the network, finding that more inhibition resulted in earlier peaks in the neural network activity.

Strengths:

(1) The experimental paradigm introduced in this study is original and well-built, allowing for the study of the targeted phenomenon. The fact that relevant neural signals were found despite the absence of sensory cues proves the setup is promising, opening the way for future works, playing with different parameters: number of tones, time between tones, sequence of temporal locations complexity, sequence of events...etc.

(2) The statistical analysis was exhaustive, the authors consistently introduced controls for different conditions and alternative hypotheses, thoroughly explaining each step of the analysis as well as the choices behind them. The supplementary figures further helped understand the data and answer interrogation one might have. This comprehensive approach was well-appreciated.

(3) The use of more biologically plausible networks, compared to traditional RNNs, to model the response of subjects is a promising approach, which can give clues as to the mechanism at play, but also make predictions that can then be proven (or disproven) by future experiments.

The authors provided a work of good technical quality and reported their methods and findings transparently, making for good reproducibility and evaluation.

Weaknesses:

(1) The most glaring weakness of the paper lies in its interpretation of the different results. Conclusions are scattered around the paper, mostly unclear, and do not always make much sense with regard to the data. For example, the authors never address the absence of a peak before the first temporal location: why would subjects not "suppress" noise before the first temporal location given its (strong) predictability? Moreover, they immediately assume a functional role for the neural signature they found, as well as a direct link between the mechanisms at play in their RNN and the human brain, thus jumping to hasty and unreliable conclusions. The authors seemed to have a strong bias towards a hypothesis (predictive gain control) and tried to fit their data into it.

- The authors cited very few relevant papers on related fields, notably on omission, and therefore did not build efficiently on previous works (e.g., Yabe, Raij, Schröger, Bekinschtein, Chait, Auksztulewicz...). Moreover, at several points in the paper, they make choices about their analysis or model without proper justification or cited sources, even when explicitly pointing to the existence of research supporting said choices.

- Only a single electrode (out of 64) was used (Cz) to carry out every analysis. Without proper justification, this choice could be misinterpreted. Moreover, adopting instead a multivariate approach (incorporating all channels) would give more strength to the paper.

- Overall, the observed electrophysiological results could be more simply explained by a mechanism akin to a go/no-go (a tone/no-tone) or omission response happening after each temporal location, as subjects have learned when to make that inference. The delay of the response with regards to temporal location would change due to error accumulation in time perception, rather than "the anticipation of the first temporal location facilitating the anticipation of the second", which makes little sense. Moreover, a response in Cz could be expected.

- As for the results of RNN, not only is the analogy with actual neurophysiological activity limited, both in principle (simple E/I dynamics) and in implementation (inference is only done at the end of each trial), but the authors do not address the activity before the first temporal location, which is a major difference with human data. Their assumption that both RNN and cluster 234 are functionally related to gain control is thus further flawed. Moreover, the analysis of the RNN is lacking, for example, the authors did not compare false positive/negative of different delays, or analyzed Wout.

- The phrasing and introduction of the paper are misleading, as confusion can arise between predicting a sequence of events (several events in a row) and predicting a single event appearing at different potential locations. It should be clarified that the paper does not address sequences of events at any point.

It seems the authors already drew their conclusion beforehand and fit the data to match this bias. As such, the interpretation of the data is messy, flawed, and often hasty, drawing erroneous conclusions and parallels.

Overall, the manuscript is of good technical quality and communicated results very transparently, but the authors seem to have a strong confirmation bias towards temporal anticipation and gain control, thus leading to flawed interpretations.

---

## [Referee Report · Reviewer #2 (Public Review)]

Summary:

In this study, the authors designed an EEG experiment to investigate how listeners use temporal structure to optimise sensory detection. Listeners heard 2 seconds of noise and had to detect a faint tone in one of 3 temporal locations (equally spaced in time). In a minority of trials, no tone was presented. Focussing on these 'no tone' trials, the authors show that the EEG 'temporally tracks' the expected tone locations. This temporal tracking behaviour is also shown in a recurrent neural network trained on the same task. The authors interpret these findings as evidence of neural gain control in the service of sequential temporal anticipation.

Strengths:

The study uses an elegant experimental design and sophisticated EEG analyses. It is striking how clear the neural signatures are (of sequential expectation in the absence of sensory input). A further strength is the use of neural network modelling to elucidate the possible neural computations.

Weaknesses:

My first major comment concerns the theoretical implications of the study. An account based on gain control and temporal anticipation seems highly plausible. But are there other plausible accounts that the current data argue against? Or are there specific versions of gain control / temporal anticipation theories that the data supports and others that the data doesn't support? To develop the manuscript, I think the authors could relate their results in a more specific way to existing accounts, outlining not only what accounts their results favor but also which accounts their data falsify. In doing so I think the study will have a stronger influence on shaping the field.

My second major comment concerns the consistent lag that is observed between tone location and neural/model responses. This would seem to be inconsistent with an anticipation account, which would instead predict zero or a negative lag. This should be discussed. While I agree the decrease in response magnitude that occurs with tone location is inconsistent with expectation violation, the positive lag that is observed seems more consistent with expectation violation than temporal anticipation/gain control.

My third major comment is a suggestion to present some further analyses that I think will be informative. First is reporting more extensively the ERP results. This currently appears in one of the panels but there are no statistical tests reported in the main text and only the tone present data is shown. Given that expectation violation has been observed most consistently with ERPs, is there evidence of this in the 'no tone' trials and if so, does it correlate over participants with the power modulation effect or rate of false alarms? Doing this analysis will possibly be informative for assessing the plausibility of different functional accounts of the data e.g. expectation violation/prediction error. My second suggestion is to report the tone present trial data. When the tone is for example presented in the first location, does the response during tone locations 2 and 3 get suppressed? And does the same occur in the neural network model? If so, this would speak to a highly dynamic form of gain control (if the gain control account is correct).

---

## [Referee Report · Reviewer #3 (Public Review)]

Summary:

The study designs an EEG experiment to study how the brain better detects targets by exploiting information about when the target may appear. The study finds that the power fluctuations of alpha and beta oscillations can indicate the time intervals in which the target may appear. Furthermore, a RNN trained on the same task can also exploit such temporal information to better detect targets at the expected time intervals.

Strengths:

(1) The design of the experiment is elegant.

(2) The EEG analysis approach is highly advanced.

(3) The study combines human EEG experiments and computational modeling to address potential computational neural mechanisms.

Weaknesses:

The RNN is used both for modeling, which is commendable, and for simulating new psychophysics experiments, which can be problematic. In other words, it is very dangerous to predict human performance in a novel condition using RNN and assume that prediction is the same as the actual human performance. Comparing the RNN performance in two different noise conditions cannot directly "suggest that the 2 Hz neural modulation observed in Corrected Cluster 234 served to enhance sensory sensitivity to the target tone at the anticipated temporal locations, while selectively suppressing sensory noise during irrelevant noise periods." Here, much stronger evidence is to actually do the behavioral tests in two noise conditions in humans, but even that behavioral experiment cannot directly indicate the function of a neural response. In other words, the conclusion "additional analyses and perturbations on the RNNs indicated that the neural power modulations in the alpha-beta band resulted from selective suppression of irrelevant noise periods and heightened sensitivity to anticipated temporal locations" is not supported. The model does not have alpha or beta oscillations at all, which is OK, but directly concluding the function of alpha/beta oscillations based on the behavior of a model that does not have these oscillations is not appropriate.

Relatedly, better detection of a target may reflect a change either in sensory processing or in decision-making, while the second possibility seems to be ignored.

The results section has a lot of discussions, which should be moved to the discussion section.